# The impact of experimental designs & system sloppiness on the personalisation process: A cardiovascular perspective

**Harry Saxton**[1,2*], **Daniel J. Taylor**[2,3,4], **Grace Faulkner**[2,3,4], **Ian Halliday**[2,3], **Tom Newman**[2,3,4,5], **Torsten Schenkel**[5,6], **Paul D. Morris**[2,3,4,5], **Richard H. Clayton**[1,2], **Xu Xu**[1,2]

**1** School of Computer Science, University of Sheffield, Sheffield, United Kingdom, **2** Insigneo Institute for *in-silico* Medicine, University of Sheffield, Sheffield, United Kingdom, **3** Division of Clinical Medicine, School of Medicine and Population Health, University of Sheffield, Sheffield, United Kingdom, **4** NIHR Sheffield Biomedical Research Centre, Sheffield Teaching Hospitals NHS Foundation Trust, Sheffield, United Kingdom, **5** Department of Cardiology and Cardiothoracic Surgery, Sheffield Teaching Hospitals NHS Trust, Sheffield, United Kingdom, **6** Department of Engineering and Mathematics, Sheffield Hallam University, Sheffield, United Kingdom

\* h.saxton@sheffield.ac.uk

**Data availability statement:** All code used to generate the findings in this study can be found

## Abstract

To employ a reduced-order cardiovascular model as a digital twin for personalised medicine, it is essential to understand how uncertainties in the model's input parameters affect its outputs. The aim is to identify a set of input parameters that can serve as clinical biomarkers, providing insight into a patient's physiological state. Given the challenge of finding useful clinical data, careful consideration must be given to the experimental design used to acquire patient-specific input parameters. Model sloppiness—where numerous parameter combinations have minimal impact on model predictions, whilst only a few parameters significantly influence outcomes—is a critical concept in this context. In this paper, we conduct the first quantification of a cardiovascular system's sloppiness to elucidate the structure of the input parameter space. By utilising Sobol indices and examining various synthetic cardiovascular measures with increasing invasiveness, we uncover how the personalisation process and the cardiovascular system's sloppiness are contingent upon the chosen experimental design. Our findings reveal that continuous clinical measures induce system sloppiness and increase the number of personalisable biomarkers, whereas discrete clinical measurements produce a non-sloppy system with a reduced number of biomarkers. This study underscores the necessity for careful consideration of available clinical data as differing measurement sets can significantly impact model personalisation.

## Introduction

The concept of digital twin (DT) originates in the 1960s with NASA creating a virtual representation in the Apollo 13 moon exploration mission. There are now many definitions of DT and one comprehensive definition is "a set of virtual information constructs that mimics the

at the Zenodo link
https://doi.org/10.5281/zenodo.15332355.

**Funding:** The author(s) received no specific
funding for this work.

structure, context and behaviour of an individual or unique physical asset, which is dynamically updated with data from its physical twin throughout its life-cycle and that ultimately informs decisions that realise value" [1]. In the realm of medicine, the potential of a DT is profound, particularly in enhancing patient care and outcomes. In the context of healthcare, a regularly updated digital representation of an individual's anatomy, physiology or diseases holds immense promise. It could empower healthcare professionals to simulate and predict a patient's disease trajectory enabling intervention and treatment to be delivered in a timely and effective way [2].

Notably, in cardiology, the adoption of heart and circulatory DTs has gradually gained momentum and trust within the clinical community, evidenced by several proof-of-concept studies [3–5]. Traditionally, clinical diagnosis and patient trajectories in cardiology rely heavily on a clinician's expertise and population-based averages [6]. However, the emergence of DTs in cardiology signifies a shift towards a more personalised approach. These DTs integrate mechanistic (physics-based) models, grounded in physiological understanding of the heart, human circulation, and related physiological processes such as baroregulation [7,8], with dynamic clinical data collected over time or immediate data available in a clinical setting [9]. This integration enables the DT tool to provide tailored predictions and assist in clinical diagnosis, catering for the unique characteristics of each patient. Virtual representations of a patient's full cardiovascular health in differing states are referred to as their "physiological envelope" [10].

Clearly, the choice of mechanistic model utilised for a cardiovascular DT is vital to ensure the correct set of physiological relevance while also maintaining some set of clinical interpretability. Lumped parameter models (LPM) offer a unique ability to examine both cardiac function and global haemodynamics. LPM provide a simple approach in which all the main characteristics of the blood flow (i.e. blood pressures, flows and volumes) are captured. Typically, an LPM is constructed of a heart chamber (acting as a blood pump), a presentation of the mechanical nature of heart valves and a series of elements representing the various vascular networks in which blood can be transported through the body. This class of model is usually represented as a system of differential algebraic equations; the size of which depends on the complexity of the system investigated (full body circulation or anatomically detailed models of specific vessels) [11].

Each LPM or compartment can be represented as a combination of resistors, capacitors and inductors which are parameterised by numerical values $R$, $C$ and $L$, respectively. For a generic vessel or organ located in a larger circulation network, $R$, $C$ and $L$ represent haemodynamic dissipation, vessel distensibility and the inertial effects of the blood flow, respectively [12]. Along with the input parameters of the heart chambers and valves, these parameters form a set of clinical biomarkers, which when personalised to a patient, by integrating patient-specific clinical data, provide the insight that a cardiovascular DT aims to achieve [13].

Useful clinical data are scarce resources, thus the requirement to identify and choose what data are needed to personalise a LPM (in order to create a useful cardiac DT) is a complex one [14]. Within a clinical setting, there are often a range of both continuous and discrete measurement data. But the process of obtaining insightful and diagnostically useful clinical data often requires a series of invasive tests being conducted on the patient. With any data collected (e.g., blood pressure, flow and volume for each compartment), one then generates a series of clinical metrics: ejection fraction [15], to quantify heart failure; pulse pressure [16], to diagnose arterial stiffening; maximum blood velocity [17], to evaluate heart valve stenosis; cardiac output [18], to measure overall heart health and the observation of various clinical time series waveforms [19]. These metrics can then be amalgamated into a DT, enriching the

model's predictions with additional detail and validity. However, given the plethora of available clinical tests, each carrying its own risks to patients, determining which metrics are indispensable in creating a faithful virtual representation of a patient becomes a challenging task. Each set of measurements collected and utilised to perform DT related tasks is denoted an experimental design [20].

The integration of clinical data into an LPM to form a DT, is a task denoted 'the personalisation process' (or 'model personalisation' or 'model calibration'). Mathematically, this is also known as the 'inverse problem' [21]. One can think of the solution to the personalisation process as an input parameter set that locates the global minimum of a response surface, spanned by the combination of input parameters of the mechanistic model and the available clinical measurements. Thus, we obtain a set of unique clinical biomarkers [22], i.e., we have found a point in the input parameter space such that the outputs of our mechanistic model most closely match the clinical measurements of a patient. This is the point in the input parameter space which describes a patient's patho/physiological state. However, calibration alone is insufficient to establish model validity. Rigorous validation across multiple operational points beyond the calibration condition is essential to ensure the DT will behave as the physical entity is extended to the broader patho/physiological state space. Without such a validation, one cannot distinguish between a genuinely predictive model and one that merely exhibits over-fitting to a single calibration point. This knowledge is vital as a DT must update this personalisation point every time new information is received. Therefore, the validation serves as a pillar of DT creation.

Despite progress, there are still many open questions surrounding the personalisation process, which we distil as explicit questions below:

1. What clinical data must be acquired *in-vivo* to obtain insightful, patient-specific biomarkers?
2. Does the set of biomarkers obtained remain consistent in the presence of new and varying experimental designs?
3. What is the computational cost associated with finding the solution of the personalisation process under different experimental designs?
4. Should DTs be built to encapsulate a patient's 'physiological envelope' or should DTs be targeted to specific conditions?
5. What are the best practices involved in model personalisation under uncertainty?

This study investigates to what extent the above questions can be answered. We aim to provide a fundamental understanding of the sloppiness present within such models before being able to deploy a functioning DT in future work. Before proceeding, it is important to note that all investigations in this work are conducted with forward generated model data, in order to understand and extract clinical biomarkers from the model in an ideal setting (i.e. to eliminate any confounding effects of noise in clinical data). The synthetic data generated from our model here are guided by clinical practice and reflect the type and key features of data obtained in clinic. The investigations performed here reflect the common clinical pathways that are taken within the hospital. Without this critical, off-line investigation, misleading parameter selection from inappropriate experimental design which could then lead to ill-informed clinical decisions.

The structure of this paper is as follows. In section background, we: (i) review relevant literature, (ii) introduce concepts germane to the personalisation process in both extraction and optimisation of clinical biomarkers, (iii) detail the position this type of investigation has in the personalisation process and (iv) summarise the principal contributions of this work. In

section methods, the mathematical detail is provided for each tool used for model analysis (Sobol indices, identifiability analysis and sloppy analysis). We also present the various experimental designs utilised within this work. Section results declares our results from different computational experiments. Discussion of the impact of varying experimental designs on the personalisation process is given in section discussion.

## Foundations and state-of-the-art

Model personalisation is synonymous with the base concepts of input parameter sensitivity, identifiability and sloppiness [23]. From the discussion in section introduction, it is prudent to review terminologies, prior art and state how this work will provide a novel insight into the study of model personalisation.

### Terminologies

When attempting to personalise a cardiovascular model, it is important to discuss the prerequisite properties corresponding to input parameter influence (sensitivity), uniqueness (identifiability) and response surface structure (sloppiness).

**Sensitivity.** An input parameter's effect needs to be influential on the output response surface, if this is the case an input parameter is regarded as **sensitive** [24], i.e., a change in the input parameter space causes a detectable change on the desired output. Thus, the said sensitive input parameter may serve as a clinically insightful biomarker for personalisation (in the creation of a digital twin), due to the ease of capturing the biomarkers' effects in clinical outputs. One can distinguish locally and globally sensitive input parameters, with respect to the measurements. Locally sensitive input parameters are those eliciting the steepest gradient in the output about the model base operating point [25]. Globally sensitive input parameters are potential bio-markers which operate within a physiologically realistic value range. Input parameters are said to be most globally sensitive when they cause the greatest influence on the outputs, for the prescribed parameter ranges [26]. Different methods exist to calculate the sensitivity of input parameters, with the most common being the variance based methods, which we adopt in this work (see section sensitivity analysis). The personalisation process and the use of cardiovascular DTs is a global process, because we need a virtual representation of patients in a range of physiological and pathophysiological conditions. Thus, global sensitivity analysis presents itself as an insightful tool in the search for of clinical biomarkers.

**Identifiability.** Personalisation of models now entails the pursuit of an identifiable model and identifiable input parameters (an optimal subset, denoted as clinical bio-markers). The analysis of identifiability in a cardiovascular system model requires three distinct examinations: structural, sensitivity-based, and practical identifiability. Structural identifiability (theoretical) assumes abundant and noise-free target output data, rendering a model's structural identifiability largely academic in clinical terms. However, this assumption overlooks the possibility that inability to identify input parameters may stem from the model's structure rather than data issues [27]. Naturally, if a model lacks structural identifiability, practical attempts at its utilisation are inherently limited. Sensitivity-based identifiability analysis involves the identification of sensitive and orthogonal input parameters, under synthetic data generated by the model, to ascertain which input parameters are identifiable under ideal circumstances [28]. Practical identifiability analysis takes into account of the quality of patient data, where noise and sampling rates may impact the identification of unique input parameters [29]. For complete personalisation, each stage must be executed sequentially. Within this work, we examine the identifiability of input parameters through their average influence across output space.

**System sloppiness.** System sloppiness is a term used to characterise the structure of the input parameter space [30]. As discussed above, the main aim in many areas of systems biology is to optimise a dynamical system's input parameters to available experimental data. This is normally performed by minimising a cost function, to obtain a point in the input parameter space corresponding to a global minimum of the cost function $J$ [31], of the form

$$J(\underline{p}) = \sum_i (y_i(\underline{p}, t) - y(t)_i^e)^2,$$

where $\underline{p}$ is the input parameter set, $y_i^e$ represents the $i$th experimental measurement available and $y_i(\underline{p}, t)$ represents the $i$th dynamical system output (obtained from the model), which the experimental data are compared against.

Consider an example dynamical system with two input parameters $p1$ and $p2$. When optimising such a system to experimental data, contour plots displaying the closeness of fit are generated as in Fig 1. Here we see that moving up (in the direction of $p2$) and left in the parameter space rapidly changes the value of the cost function, i.e., indicating how good a fit is obtained by a specific value of $p2$. This direction is denoted a stiff direction in the input parameter space. Conversely, if one was to travel up and right (in the direction of $p1$), one could visit a range of $p1$ values without incurring changes of the cost function values. This means the manifold generated by $p1$ is largely linear whereas the one generated by $p2$ has steep gradients leading to a unique global minimum. Thus the sloppy direction controlled by $p1$ would not make for a good biomarker to calibrate a model due to the limited impact of $p1$ to the selection of model outputs. The converse is true for $p2$.

However, most models in systems biology and in cardiovascular modelling cannot be visualised through a two dimensional contour map. Mathematical sloppiness analysis of our cardiovascular models can be found in section System Sloppiness.

The final stage of personalisation is the optimal estimation of the selected input parameters, fitted to patient-specific clinical data. This estimation may be iteratively updated within a DT as new information and data from the patient become available. Before the optimisation step takes place, the quantification of the stiffness/sloppiness of the system's parameter space provides insight into the complexity of the system, and in turn facilitates the choice of

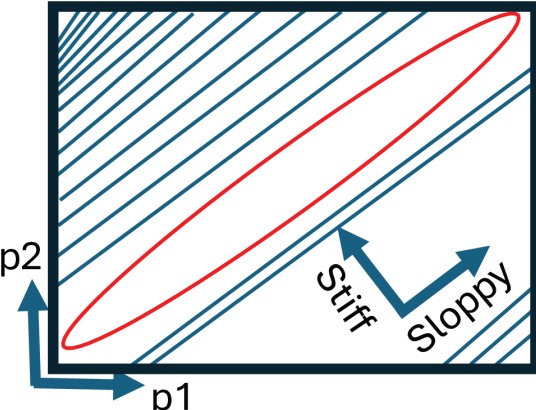

**Fig 1. A two dimensional sloppy model representation: A two dimensional contour plot displayed as a blue curve, with the minimum contour value displayed in red for input parameters $p1$ and $p2$.** Moving up and left would lead to rapid changes in the contour where as moving up and right would lead to slow changes.

an appropriate optimisation routine (e.g. gradient descent, particle swarm genetic algorithms, unscented Kalman filter etc.) [32,33]. A sloppy system tends to have elongated or even flat valleys around the minimum, meaning that the minimum is not well-defined. Of course, when applying this method to patients whose measurements are noisy, the sloppiness of the model can vary significantly due to influence from the noise.

The related concepts of identifiability and sloppiness provide different but insightful information about the personalisation process [34]. Identifiability is a binary situation, whilst sloppiness quantifies the difficulty associated with obtaining precise identifiable input parameters. The sloppiness analysis of a model can either distinguish stiff and sloppy regions of the input parameter space, or show that the whole system under investigation can be regarded as sloppy. Note, most system biology models belong to the latter category [35,36]. Within this work, we examine the sloppiness associated with the sensitivity matrices, which are defined by input parameter effects on the chosen outputs. Therefore, we can also establish a secondary aim of investigating the effects of differing experimental design on a cardiovascular system sloppiness.

## Relevant literature

Cardiovascular model personalisation has been attempted in many clinically important areas such as, congenital heart disease, fetal circulation or whole heart multi-scale modelling [37–39]. In the majority of works, standard optimisation routines are used to obtain a set of input parameters which are representative of the experimental data [40–42]. Outside of the standard optimisation routines, data assimilation methods, namely ensemble and unscented Kalman filters, have developed traction as an efficient way to estimate patient specific input parameters, at a reduced cost, compared to the optimisation methods [33,43,44]. Another area which attracted research community's attention recently is the utilisation of sensitivity analysis to guide the search and selection of an optimal input parameter subset for simpler and more efficient parameter estimations. Colunga et al. [45] applied this technique to incorporate invasive right heart data to obtain the personalisation of a model of pulmonary hypertension with 25 parameters. Where as Strocchi et al. [46] applied global sensitivity analysis to a 117-parameter cell-to-organ lumped parameter 4-chamber heart representation, reducing the model down to 45 personalisable parameters. Schafer et al. [47] examined how the sensitivities in a 1D model of the carotid artery change with respect to age and sex, highlighting how the input parameters for personalisation do not remain constant.

As discussed above, it is also important to understand the identifiability of input parameters, because this provides reassurance that any optimised input parameters are unique to a patient. Casas et al. [48] performed a profile likelihood analysis of a LPM to personalise flow in the systemic circulation. In comparison, other researchers such as Pironet et al., [49,50] performed a structural and sensitivity identifiability analysis on a LPM to highlight what outputs would be required to obtain unique input parameters. In addition, there have been developments of experimental approaches from Marquis et al., de Bournonville et al. and Sala et al., who used invasive patient data to make good first estimates of model input parameters before optimisation which ensured more input parameters within the model are identifiable [51–53].

Another popular approach was to optimise input parameters in an iterative manner [54]. Bjordalsbakke et al. [55] applied an iterative step-wise reduction scheme in which, guided by sensitivity analysis, they began to optimise a group of parameters with increased number, each time with different cost functions to examine the closeness of fit. Bjordalsbakke [55] found that cost functions constructed from waveform data, as opposed to common clinical metrics, produced the smallest errors. Hann et al. [56], took a similarly structured approach in

reducing the number of available outputs to reduce the complexity associated with the model. They demonstrated that differing measurement sets consisting of continuous and discrete measurements allowed for each subsection of the model to be optimised with minimal error. The impact of varying outputs have also been examined by Eck et al. [57] through uncertainty quantification in the arterial wall models and concluded that continuous time series Sobol indices gave a more insightful look into the process.

All studies above indirectly examined the impact of changing outputs on obtaining personalisable input parameter sets. However this was not the primary objective of their investigations. The most notable publication to date which investigated the impact of experimental design on a cardiovascular model was that of Colebank et al. [58]. They studied the impact of 4 different experimental designs on the ventricular function and found easily identifying biomarkers of the ventricular function, when a practical identifiability analysis is performed (including continuous data from both the left and right side of the heart).

Another aim of our study is to reveal and analyse the impact of experimental designs on system sloppiness. Sloppiness is a property which has been known in system biology models for over a decade [35]. Most studies have been focused on pharmacokinetic models [59] and examinations of the route cause of sloppiness. More recently, a formal definition of sloppiness was given and the concept was used to obtain a minimum set of outputs to ensure identifiability [60,61]. The impact of experimental designs on system sloppiness has been understood through the lens of model identifiability (i.e., varying the model and the data shown to the model can induce different intensity of model sloppiness which in turn impacts the overall identifiability of the model parameters) [62–64]. In terms of cardiovascular modelling, sloppy analysis has been applied to electrophysiology modelling with the focus on calcium and potassium channel modelling. Whitterker et al. [65], utilised sloppy analysis to provide a method to simplify complex models of ion channels that improves parameter identifiability which will aid in future development in voltage-gated ion channels. Sloppy analysis has been applied to other classes of biochemical models [66,67]. For example, Bravo et al. [36] applied sloppy analysis to a Bayesian model of an electrophysiological process to highlight how the identification of stiff parameter combinations made the model personalisation much simpler. As far as we are aware, sloppy analysis of a mechanical LPM of the cardiovascular system has not been performed, nor has the impact of varying experimental designs on sloppiness been studied.

## Rationale and contributions

DTs in cardiovascular medicine offer profound promise in improving patient care. In order to advance the application of DTs in clinic, further study into the impact of experimental designs on the personalisation process must be understood. We investigate the impact of varying clinical metrics, both continuous and discrete, in an ideal scenario (without the bias of measurement noise). The main contributions of the work are:

1. **Stability of identifiable parameters:** Through varying experimental designs, we investigate changes in the identifiability ranking of input parameters.
2. **Sloppy analysis of an LPM:** We perform and report the first global sloppy analysis of a cardiovascular LPM to aid our understanding of the personalisation process.
3. **Clinical metric guidelines:** By evaluating different clinical metrics, we provide insight into the set of clinical data (and therefore measurements) needed for the effective personalisation of cardiovascular models.

By rigorously assessing the impact of varying clinical metrics and performing the first insightful sloppy analysis of a cardiovascular model, revealing the structure of the input parameter space, our work furthers the investigation and understanding needed in the personalisation of cardiovascular DTs.

## Methods

Here, we examine the methods used for analysing a lumped parameter 4-chamber cardiovascular circulation model. In section methods, we present the model, explain the computational framework and provide a full parameterisation with their initial conditions. We then detail the clinical measures utilised in this work, which map to the different measurement sets in section Clinical Measures. The global sensitivity method of Sobol indices and the related Fisher information matrix are given in section sensitivity analysis and section FIM respectively. In section average influence, we then examine an average metric for input parameter influence. We detail how to examine system sloppiness in an $n$–dimensional space in section sloppy analysis. Finally, section workflow is devoted to highlight the iterative workflow we devised to examine the effects of varying experimental design on input parameter influence and sloppiness.

### Cardiovascular model

Our LPM can be expressed in a standard state space formulation:

$$\frac{d}{dt}\underline{X}(t) = \underline{f}\left(\underline{X}(t); \underline{p}\right), \quad \underline{Y}(t) = \underline{h}(\underline{X}(t)), \tag{1}$$

in which $p$ denotes an input parameter vector, $\underline{X}$ represents the set of state variables of the system, $f$ is a function describing the system (usually this is an collection of differential algebraic equations), $\underline{h}$ is the measurement function where the forward model synthetic measurements are generated, using the computed state variables $\underline{X}$, and $\underline{Y}$ represents the measurements of interest.

The model declared in its electrical analogue form in Fig 2 is a system-set, differential algebraic equation based, electrical analogue cardiovascular model, after Comunale et al., [68], with 4 heart chambers and a representation of both the systemic and pulmonary circulations. The model was first developed to model both physiological and pathophysiological states. The state variables of each compartment are specified by its time-dependant dynamic pressure $P$ (mmHg), inlet flow $Q$ (mL/s) and volume $V$ (mL):

$$X_k(t) = \left(V_k(t), P_k(t), Q_k(t)\right), \quad k \in \{la, lv, sa, svb, sv, ra, rv, pa, pvb, pv\}, \tag{2}$$

where *la* denotes the left atrium, *lv* denotes the left ventricle, *sa* the systemic arteries, *svb* the systemic vascular bed, *sv* the systemic venous system, *ra* the right atrium, *rv* the right ventricle, *pa* the pulmonary arteries, *pvb* the pulmonary vascular bed and *pv* the pulmonary venous system, Formally, *t* is a continuous time variable.

In generic form, the equations relating to the passive compartmental state variables all take the form:

$$\frac{dV_{s,k}}{dt} = Q_k - Q_{k+1}, \quad \frac{dP_k}{dt} = \frac{1}{C_k}(Q_k - Q_{k+1}), \quad Q_k = \frac{P_{k-1} - P_k}{R_k}. \tag{3}$$

Above, the subscripts $(k-1), k, (k+1)$ respectively represent the proximal, present and distal system compartments; $V_{s,k}$(mL) denotes the circulating (stressed) volume [70] and $C_k$

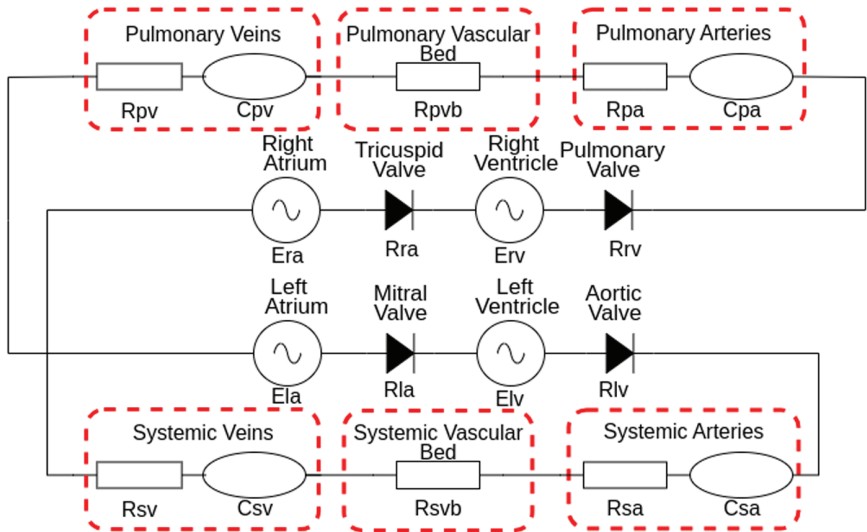

**Fig 2. Lumped 4-chamber cardiovascular model: Our state-space cardiovascular model, first introduced by Comunale et al. [68].** Both left and right atria and ventricles are represented by the Shi double cosine model [69]. Heart valves are assumed to have Ohmic behaviour allowing no back flow. The systemic and pulmonary circulations are represented by a CRRCR combination with all parameters are given in Table 1.

(ml/mmHg) and $R_k$ (mmHgs/mL) denote compartmental compliance and the Ohmic resistance between compartments $k$, $(k + 1)$. Flow in and out of the active heart chambers are controlled by Ohmic diode valves:

$$Q_k = \begin{cases} \frac{P_{k-1} - P_k}{R_{val}}, & P_{k-1} > P_k, \\ 0 & P_{k-1} \leq P_k, \end{cases} \tag{4}$$

where $R_{val} = (R_{lv}, R_{la}, R_{rv}, R_{ra})$ representing the resistances across aortic, mitral, pulmonary and tricuspid valves.

The active heart chambers can be represented by time varying elastances $E_k(t)$ (mmHg/ml), which determines the change in pressure for a given change in the volume [70]:

$$E_k(t) = \frac{P_k(t)}{V_k(t) - V_{k,0}} = \frac{P_k(t)}{V_{k,s}(t)}, \quad k \in \{lv, rv, la, ra\} \tag{5}$$

where $V_{k,0}$ & $V_{k,s}(t)$ represent the chamber unstressed and stressed volumes for the two ventricles and atria. $E_k(t)$ is written following [69]:

$$\tilde{t} = \text{Mod}(t + (1 - E_{k,shift})\tau, \tau)$$
$$E_k(\tilde{t}) = (E_{k,max} - E_{k,min}) \cdot e(\tilde{t}) + E_{k,min},$$
$$e(\tilde{t}) = \begin{cases} \frac{1}{2}\left[1 - \cos(\frac{\pi\tilde{t}}{\tau_{k,es}})\right], & 0 \leq \tilde{t} < \tau_{k,es}, \\ \frac{1}{2}\left[1 + \cos(\frac{\pi(\tilde{t}-\tau_{k,es})}{\tau_{k,ep}-\tau_{k,es}})\right], & \tau_{k,es} \leq \tilde{t} < \tau_{k,ep}, \\ 0, & \tau_{k,ep} \leq \tilde{t} < \tau, \end{cases} \tag{6}$$

where $e(t; \tau_{k,es}, \tau_{k,ep}, E_{k,shift})$ is the activation function for the heart chamber and is parameterised by the end systolic and end pulse timing parameters $\tau_{k,es}$ and $\tau_{k,ep}$ respectively.

Table 1 provides the parameterisation for the model to produce physiological results for the lumped parameter 4-chamber model in which the input parameter space has a dimensionality of 36. To construct the model, the cardiovascular LPM Julia package CirculatorySystemModels.jl [71] is utilised which reduces the system to 8 ordinary differential equations in volume with the initial volumes also given in Table 1. The model is solved utilising the Vern7 [72] algorithm with tolerances of $1e^{-6}$. For 30 cardiac cycles the model computes in 0.064s.

## Clinical measures

When varying the experimental design, we generate synthetic data based on medically accurate measures utilised in the diagnoses of cardiovascular diseases. This enables us to perform such a large computational investigation, and at the same time, to follow the standard clinical pathway through an iterative scheme. To investigate the effect of experimental design, we devise an additive algorithm: each time we move to the next measurement set, the new measurement is added to the previous output set and therefore defining the new output space of an increased dimension, for the analysis of input parameter effects. Practically, the below

**Table 1. Input parameters for the lumped parameter 4-chamber model: Each input parameter is displayed along with the respective units and valves. Here we fix the heart period cycle to $\tau = 0.81(s)$.**

| Heart Parameters | | | | | |
|---|---|---|---|---|---|
| **Parameter Name** | **Symbol** | **LV** | **RV** | **LA** | **RA** |
| Maximal Elastance [$mmHg/ml$] | $E_{max}$ | 2.8 | 0.45 | 0.13 | 0.09 |
| Minimal Elastance [$mmHg/ml$] | $E_{min}$ | 0.07 | 0.035 | 0.09 | 0.045 |
| Unstressed Volume [$ml$] | $V_0$ | 20 | 30 | 3 | 7 |
| End Systolic Time [$s$] | $\tau_{es}$ | $0.269\tau$ | $0.269\tau$ | $0.11\tau$ | $0.11\tau$ |
| End Diastolic Time [$s$] | $\tau_{ep}$ | $0.452\tau$ | $0.452\tau$ | $0.18\tau$ | $0.18\tau$ |
| Atrial Activation Time [$s$] | $E_{shift}$ | 0 | 0 | $0.85\tau$ | $0.85\tau$ |
| Valve Resistance [$mmHg \cdot s/ml$] | $R_{val}$ | 0.01 | 0.01 | 0.005 | 0.005 |
| Circulation Parameters | | | Initial Volume Values | | |
| Resistance Systemic Arteries [$mmHg \cdot s/ml$] | $R_{sa}$ | 0.0448 | Initial Volume Systemic Arteries [$ml$] | $V_{sa,0}$ | 98.3 |
| Resistance Systemic Vascular Bed [$mmHg \cdot s/ml$] | $R_{svb}$ | 0.824 | Initial Volume Systemic Veins [$ml$] | $V_{sv,0}$ | 117.996 |
| Resistance Systemic Veins [$mmHg \cdot s/ml$] | $R_{sv}$ | 0.0269 | Initial Volume Pulmonary Arteries [$ml$] | $V_{pa,0}$ | 100.5 |
| Resistance Pulmonary Arteries [$mmHg \cdot s/ml$] | $R_{pa}$ | 0.003 | Initial Volume Pulmonary Veins [$ml$] | $V_{pv,0}$ | 126.4 |
| Resistance Pulmonary Vascular Bed [$mmHg \cdot s/ml$] | $R_{pvb}$ | 0.0552 | Initial Volume Left Ventricle [$ml$] | $V_{lv,0}$ | 149.6 |
| Resistance Pulmonary Veins [$mmHg \cdot s/ml$] | $R_{pv}$ | 0.0018 | Initial Volume Right Ventricle [$ml$] | $V_{rv,0}$ | 189.2 |
| Compliance Systemic Arteries [$ml/mmHg$] | $C_{sa}$ | 0.983 | Initial Volume Left Atrium [$ml$] | $V_{la,0}$ | 71 |
| Compliance Systemic Veins [$ml/mmHg$] | $C_{sv}$ | 29.499 | Initial Volume Right Atrium [$ml$] | $V_{ra,0}$ | 67 |
| Compliance Pulmonary Arteries [$ml/mmHg$] | $C_{pa}$ | 6.7 | | | |
| Compliance Pulmonary Veins [$ml/mmHg$] | $C_{pv}$ | 15.8 | | | |

model outputs represent the conventional medical tests which a patient may be subjected to, with increasing invasiveness to assess their cardiovascular health. While we do not target a specific health condition in this work, the increasing output sets represent further and deepening knowledge about a patient's physiological envelope.

In Tables 2, 3, and 4, we display the measurement sets for the cases of discrete, continuous and mixed measurements. These will then be utilised to perform the variety of investigations as described below and pictured graphically in Fig 3.

**Discrete measurements.** In the discrete case in Table 2, we utilise only single point metrics. These metrics can be obtained through just 3 clinical tests:

1. Blood Pressure (BP): This can be readily obtained through a sphygmomanometer reading while a patient is in hospital. In our chosen model, this measurement is obtained by calculating $\frac{\text{Max}(P_{sa})}{\text{Min}(P_{sa})}$ and corresponds to set 1.
2. Ejection Fraction (EF): This can be obtained through an echocardiogram. In our model, we calculate EF for the left and right ventricle then the left and right atria as $\frac{\text{Max}(V_i)-\text{Min}(V_i)}{\text{Min}(V_i)}$. For $i = lv, rv, la, ra$ this corresponds to sets 2A, 2B, 2C and 2D.
3. Max($Q_i$) - Maximum flow: This could be obtained from either an echocardiogram or a cardiac MRI. This is calculated for the systemic arteries, pulmonary arteries, aortic valve, mitral valve, pulmonary valve and tricuspid valve.

To highlight the additive process of the experiment in the discrete setting, for example, the full output set for 3A is defined as follows:

$$\text{Set 3A} = \big(BP, EF_{lv}, EF_{rv}, EF_{la}, EF_{ra}, \text{Max}(Q_s)\big).$$

**Continuous measurements.** For the continuous measurements displayed in Table 3, each continuous waveform obtained is made up of 150 time points. This metrics can be obtained though 4 clinical metrics below.

1. $Q_i$ - Flow rate: This can be obtained through a Doppler ultrasound, for the systemic, pulmonary, aortic valve, mitral valve, pulmonary valve and tricuspid valve.
2. $V_i$ - Chamber volume: can be obtained through a cardiac MRI, for the two ventricles and two atria.
3. $P_{lv}, P_{sa}$ - Left heart pressures: These invasive diagnostic measurements can be obtained through catheterisation for the left ventricle and systemic artery.
4. $P_{rv}, P_{pa}, P_{ra}, P_{pv}$ - Right heart and circulation pressures: These invasive measurements can be obtained by performing Swan-Ganz catheterisation and wave form pressures

**Table 2. Table of discrete measurements:** This table presents the sequential addition of discrete measurements to the experimental design. Each pairing represents a measurement set, showing which new cardiovascular measurement is added whilst retaining all previous measurements. These discrete measurements represent single-point values extracted from the cardiac cycle, reflecting clinical metrics commonly used for cardiovascular assessment with increasing physiological depth.

| Discrete Measurement Sets | | | | | | |
|---|---|---|---|---|---|---|
| Set | 1 | 2A | 2B | 2C | 2D | |
| Measurement Added | BP | $EF_{lv}$ | $EF_{rv}$ | $EF_{la}$ | $EF_{ra}$ | |
| Set | 3A | 3B | 3C | 3D | 3E | 3F |
| Measurement Added | Max($Q_s$) | Max($Q_p$) | Max($Q_{lv}$) | Max($Q_{la}$) | Max($Q_{rv}$) | Max($Q_{ra}$) |

**Table 3. Table of continuous measurements:** This table illustrates the progression of continuous measurement sets employed in the cardiovascular model analysis. Each set incorporates a new continuous waveform measurement (consisting of 150 time points from a complete cardiac cycle) whilst preserving all previous measurements. The progression moves from non-invasive to increasingly invasive measurements, mirroring typical clinical pathways for comprehensive cardiovascular assessment.

**Continuous Measurement Sets**

| Set | 1A | 1B | 1C | 1D | 1E | 1F |
|---|---|---|---|---|---|---|
| Measurement Added | $Q_s$ | $Q_p$ | $Q_{lv}$ | $Q_{la}$ | $Q_{rv}$ | $Q_{ra}$ |
| Set | 2A | 2B | 2C | 2D | 3A | 3B |
| Measurement Added | $V_{lv}$ | $V_{rv}$ | $V_{la}$ | $V_{ra}$ | $P_{lv}$ | $P_{sa}$ |
| Set | 4A | 4B | 4C | 4D | | |
| Measurement Added | $P_{rv}$ | $P_{pa}$ | $P_{ra}$ | $P_{pv}$ | | |

**Table 4. Table of mixed measurements:** This table details the mixed measurement protocol combining both discrete and continuous cardiovascular measurements in a clinically-relevant sequence. Each set (except set 2, which replaces set 1) adds a new measurement to the cumulative experimental design, progressing from simple non-invasive assessments to comprehensive invasive monitoring. This reflects realistic diagnostic pathways where discrete measurements (single values) are often obtained before continuous waveforms (150 time points per cardiac cycle) are recorded, particularly for invasive parameters.

**Mixed Measurement Sets**

| Set | 1 | 2 | 3A | 3B | 3C | 3D |
|---|---|---|---|---|---|---|
| Measurement Added | BPN | BP | $EF_{lv}$ | $EF_{rv}$ | $EF_{la}$ | $EF_{ra}$ |
| Set | 4A | 4B | 4C | 4D | 4E | 4F |
| Measurement Added | $Q_s$ | $Q_p$ | $Q_{lv}$ | $Q_{la}$ | $Q_{rv}$ | $Q_{ra}$ |
| Set | 5A | 5B | 5C | 5D | 5E | |
| Measurement Added | $\mathrm{Max}(Q_{lv})$ | $\mathrm{Max}(Q_{la})$ | $\mathrm{Max}(Q_{rv})$ | $\mathrm{Max}(Q_{ra})$ | $V_{lv}$ | |
| Set | 5F | 5G | 5H | 6A | 6B | |
| Measurement Added | $V_{rv}$ | $V_{ra}$ | $V_{ra}$ | $P_{sv}$ | $P_{sa}$ | |
| Set | 7A | 7B | 7C | 7D | | |
| Measurement Added | $P_{rv}$ | $P_{pa}$ | $P_{ra}$ | $P_{pv}$ | | |

are collected in the right heart for the right ventricle, pulmonary artery, right atrium and pulmonary vein (which can also be seen as a surrogate for left atrial pressure).

**Mixed measurement sets.** The previous two measurement settings will reveal the difference between continuous and discrete metrics. The mixed measurement set combines both the discrete and continuous measurements but represents a standard diagnosis procedure with increasing invasiveness, i.e., in clinic, a patient would not be subject to invasive chamber pressure measurements unless deemed necessary. Apart from one additional measurement (noisy blood pressure) which will be introduced bellow, all other metrics and corresponding measurement sets are the same as the ones defined in sections Discrete and Continuous Measurements.

- BPN - noisy blood pressure: This set is added to represent the situation of a patient taking their own arterial blood pressure measurement at home, with noisy reading due to

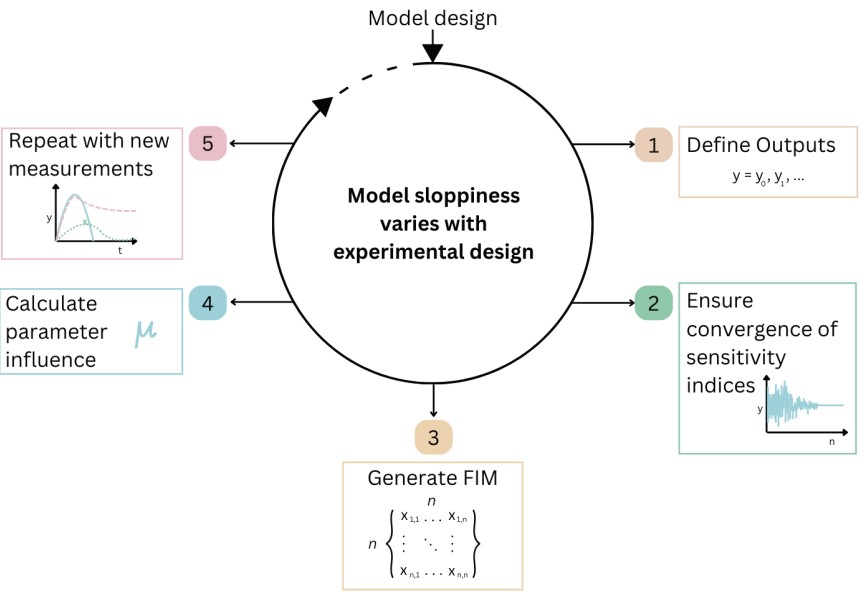

**Fig 3. Experimental design workflow: The process for how to analyse the input parameter influence and sloppiness in the presence of changing experimental designs.**

potential human error and lower device accuracy. The analysis of noisy output data will provide insight into how global sensitivity indices alter in the presence of noise.

BPN is calculated as

$$\text{BPN} = \frac{\text{Max}(P_{sa})}{\text{Min}(P_{sa})} \times (1 + \epsilon), \quad \epsilon \sim N(0, 0.1).$$

In this setting, set 2 represents an arterial blood pressure measurement obtained in hospital and is assumed to not be subject to noise. Slightly different to the measurement sets introduced in sections Discrete and Continuos Measurements, here set 2 will replace set 1 instead of adding to it and set 2 will be used in all increasing measurement sets for the later sets.

## Sobol indices

Given a model of the form in eqn 1 with $Y$ (a continuous or discrete output), a variance based first order or total order effect can be calculated for a generic input factor $p_i$. $p_i^c$ denotes the complementary set, i.e., all other model inputs excluding $p_i$. Performing a Sobol analysis provides the quantification of the input parameter effect against a specific output [73]. Both the first and total order sensitivity indices return a matrix of the form:

$$S = S_{j,i}, \quad j = 1, ..., m; \quad i = 1, ..., n, \tag{7}$$

where $n$ and $m$ represent the number of input parameters and output measurements, respectively. In this work, we handle a fixed high-dimensional input parameter space ($n = 36$). However, when the experimental design changes, $m$ varies and also the output measurement space. For example, in the discrete measurement setting, the largest value of $m$ is 11. In comparison, in the mixed measurement setting, the largest output set (set 7D) results in $m = 1609$, so the resulting sensitivity matrix is of size $S = (1609 \times 36)$.

The first and total order sensitivity indices can be written as:

$$S_{1,i}(Y) = \frac{\mathrm{Var}(\mathrm{E}(Y|p_i))}{\mathrm{Var}(Y)}, \quad S_{T,i}(Y) = \frac{\mathrm{E}(\mathrm{Var}(Y|p_i^c))}{\mathrm{Var}(Y)}, \tag{8}$$

where $S_{1,i}$, $S_{T,i}$ denote the first and total order indices' vectors for an input parameter $p_i$ against the specific output $Y$. In order to quantify the effects continuous measurements have on the calculation of sensitivity indices, we typically average this sensitivity waveform. Rather than averaging across a time range (which process regions of low variance equally to those of high variance), one seeks to expose differential sensitivities by examining variance-weighted averages:

$$TAS_i = \frac{\sum_k S_i(\underline{Y}^c(t_k))\mathrm{Var}(\underline{Y}^c(t_k))}{\sum_k \mathrm{Var}(\underline{Y}^c(t_k))}, \tag{9}$$

where $TAS_i$ is the time averaged first/total order effect of an input parameter $i$ and $\underline{Y}^c(t_k)$ represents the approximated continuous measurement at time step $k$ [57].

The sensitivity indices can be interpreted as:

$$S_{T,i} = S_i + \sum_{i \neq j} S_{ij} + \sum_{i \neq j \neq k} S_{ijk} + ...,$$

i.e., for a given input parameter $p_i$, the total order indices are the first order indices ($p_i$'s independent effects) *plus* all higher order interactions. This study utilises the total order sensitivity matrix to quantify an input parameter's full impact on the outputs. To ensure convergence, we used 75,000 samples with the Jansen estimator [74,75] with a bootstrapping sample of 1000 [76]. In this work we prescribe a uniform distribution on the inputs with the upper and lower bounds for the parameters being at $\pm 50\%$, from the values shown in Table 1, and we generate a quasi-Monte carlo Sobol sequence to improve the convergence of the Sobol indices [75].

## Fisher information

Another important matrix derived from the $(m \times n)$ sensitivity matrices is the square $(n \times n)$ Fisher information matrix (FIM) [77]:

$$\mathbf{F} = \mathbf{S}^T\mathbf{S}. \tag{10}$$

The FIM is a symmetric matrix representing the information one can extract on parameters from the model outputs (i.e., the available measurements [78]). We choose to construct the FIM from the total order Sobol indices (eqn 8) to account for the full non-linear effects which are present within the system and the impacts of the varying experimental designs as done in previous work [25,79,80].

## Average parameter influence

The sensitivity vectors derived in section Sensitivity Analysis only display the effects of an input parameter against a specific measurement. In order to obtain an input parameter's identifiability, we must assess an input parameter's influence across our chosen set of outputs. Li et al., [81], derived such a metric based upon the FIM $\mathbf{F}$ defined in eqn (10). To examine the global effect of input parameters through the FIM, we use principal component analysis (PCA) [82] where the principal components (PC) are the eigenvectors of the FIM.

Let $Q$ be the matrix of the ordered PC (eigenvectors of $F$), in which the absolute value of each element $Q_{ij}$ reflects the contribution of the $i^{th}$ parameter to the variance of the j$^{th}$ output. We follow Li et al. [81], who measure an overall effect for the $i^{th}$ parameter as:

$$E_i = \frac{\sum_{j=1}^{m} |\mu_j Q_{ij}|}{\sum_{j=1}^{m} |\mu_j|}, \tag{11}$$

where $0 \leq E_i \leq 1$ and $\mu_j$ represents the non-zero eigenvalues of $F$. This measure reflects the difficulty in determining the $i^{th}$ parameter when only a single factor is estimated. Parameter identifiability is associated with $E_i$ - the larger the value of $E_i$, the more identifiable the $i^{th}$ parameter is. We record the rank and the overall effect of every input parameter greater than 0.01, a number which has been discussed as the lowest possible value that may have the possibility of being identifiable in [25].

## Sloppiness analysis

In section System Sloppiness, we explored the concept of sloppiness by examining the contour lines of the cost function. In order to examine sloppiness in an $n$-dimensional input parameter space, we examine the eigenvalues of FIM (eqn 10). The eigenvalues of the FIM provide insight into the variation of parameters constrained by the available data. A model can be regarded as sloppy if the eigenvalues of the FIM have a uniform spacing on a log scale spread over many orders of magnitude [30,63,83,84]. On the other hand, if the FIM eigenvalues have a non-uniform distribution, the model is regarded as stiff. We can then identify stiff directions in the input parameter space which corresponds to a set of input parameters where personalisation should take place.

One can interpret this analysis as follows, the eigenvalues of the FIM represents the variation that a model parameter contributes the model outputs. In the case where a model is stiff we have a select subset of model parameters with large eigenvalues compared to their complementary set. Thus when looking to calibrate model parameters this stiff subset of input parameters denote rapidly varying directions on the response surface, thus one can more easily obtain a personalised operating point of input parameters, due to the impact of varying input parameter sets been clear. Conversely in the case of sloppy models these properties are not clear in the response surface thus choosing the correct direction for personalisation more difficult.

## Workflow

All the above sections define an iterative investigation in which we examine the effects of varying experimental designs on input parameter's influence and sloppiness. This can be encapsulated in the following steps and in Fig 3.

1. Define an output set: As shown in section Clinical Measures, we define the various sets of discrete, continuous and mixed measurements, starting from the first and simplest output set.
2. Calculate $S_T$: Form the total order sensitivity matrix for the input parameters and chosen output set.
3. Form the FIM: Utilising eqn 10, the FIM represents all the information about the parameters constrained by the specific measurement set.

4. Analyse parameter's influence and sloppiness: Using methods in section Average Parameter Influence and Sloppy Analysis, we record each input parameter's rank and influence value being greater than 0.01 and the distribution of eigenvalues.

5. Add a new measurement: Move to the next measurement set and repeat stages 2-4.

## Results

Sections Results Discrete, Continuos and Mixed detail the average input parameter influence across all outputs for varying experimental design, using the method presented in section Average Parameter Influence. The average input parameter rank and values are displayed as tables. The corresponding fig in each section displays the eigenvalues of the Fisher information matrix on a $\log_{10}$ scale, for varying measurement sets (see sections Fisher Information and Sloppy Analysis). The sections below presents results for the varying discrete, continuous and mixed measurement sets, respectively, as described in section Clinical Measures.

### Discrete measures

Table 5 displays the rank of each input parameter and their corresponding influence value calculated using eqn 11, using discrete measurements. We note, from set 1 to set 3F, the resistance of the systemic vascular bed $R_{svb}$ and the systemic arterial compliance $C_{sa}$ rank the highest, with the largest influence values, with the exception from set 2D in which $E_{shift,ra}$ ranks the most influential. This can be explained by the experimental design. The newest measurement added for set 2D was the ejection fraction of the right atrium. Although in all the other cases, it appears $R_{svb}$ and $C_{sa}$ still dominate. As more measurements are added to the experimental design, we observe more input parameters record an influence score greater than 0.01. For case 3F where there are 11 outputs, 17 input parameters are recorded with an influence score larger than 0.01. As the measurement set increases, the largest influential value decreases. In addition, as more measurements are added, although more "influential" parameters are obtained, the majority have an influence score in the range of hundredths.

The sloppiness analysis result in Fig 4 shows that the model cannot be regarded as sloppy, with a discrete output set. With an increasing output set, we observe more input parameters are regarded as stiff. Even with the largest output set, set 3F, the model still exhibits an eigenvalue spectrum of over 15 orders of magnitude.

### Continuous measures

With an increasing continuous measurement set, in Fig 6, a much higher number of influential input parameters are present, compared to the discrete case (Fig 5). Even with just a single continuous measurement of the systemic flow (Column 1A), 17 input parameters are regarded as influential. In set 4D, where there are 16 continuous output measurements, 20 input parameters are recorded as influential. Here, the ranking of influential input parameters shows much less consistency, compared to Table 5. The most influential parameters appear to loosely correlate with the latest output added to the measurement list. For example, in set 3B, the left ventricular pressure and systemic arterial pressure are new additions to the output set, and then the top ranking parameters are the minimal ventricular elastance $E_{min,lv}$ and the end pulse time for the left ventricle $\tau_{ep,lv}$. Where as in set 4C, pressures associated with the pulmonary system have just been added, then the top ranking parameters are the right atrial activation time $E_{shift,ra}$ and the minimal elastance for the right ventricle $E_{min,rv}$. Despite the change in rankings, we note that a similar set of input parameters are recorded as influential input parameters, with just minor changes when new output measurements are added. As

**Table 5. Input parameter ranking for discrete measurements: Each column displays parameters (P) and their influence values (E) calculated using Eq (11), shown in decreasing order of influence. Results are presented for progressively expanding discrete measurement sets (from Set 1 to Set 3F). Only parameters with influence values greater than 0.01 are shown.**

**Measurement Sets**

| 1 (P) | E | 2A (P) | E | 2B (P) | E | 2C (P) | E | 2D (P) | E |
|---|---|---|---|---|---|---|---|---|---|
| v $C_{sa}$ | 0.80 | $R_{svb}$ | 0.44 | $R_{svb}$ | 0.34 | $R_{svb}$ | 0.26 | $E_{shift,ra}$ | 0.24 |
| $R_{svb}$ | 0.60 | $C_{sa}$ | 0.43 | $C_{sa}$ | 0.33 | $C_{sa}$ | 0.26 | $R_{svb}$ | 0.10 |
| $\tau_{es,lv}$ | 0.04 | $E_{max,lv}$ | 0.24 | $E_{max,lv}$ | 0.18 | $E_{max,lv}$ | 0.14 | $C_{sa}$ | 0.10 |
| | | $V_{0,lv}$ | 0.04 | $V_{0,lv}$ | 0.03 | $V_{0,lv}$ | 0.03 | $E_{max,lv}$ | 0.06 |
| | | $\tau_{es,lv}$ | 0.02 | $E_{max,rv}$ | 0.02 | $E_{max,rv}$ | 0.02 | $E_{max,ra}$ | 0.04 |
| | | | | $\tau_{es,lv}$ | 0.02 | $\tau_{es,lv}$ | 0.01 | $\tau_{ep,rv}$ | 0.02 |
| | | | | | | | | $E_{min,rv}$ | 0.01 |
| | | | | | | | | $E_{max,rv}$ | 0.01 |
| | | | | | | | | $V_{0,lv}$ | 0.01 |

**Measurement Sets**

| 3A (P) | E | 3B (P) | E | 3C (P) | E | 3D (P) | E | 3E (P) | E | 3F (P) | E |
|---|---|---|---|---|---|---|---|---|---|---|---|
| $R_{svb}$ | 0.22 | $R_{svb}$ | 0.20 | $R_{svb}$ | 0.19 | $R_{svb}$ | 0.17 | $R_{svb}$ | 0.15 | $R_{svb}$ | 0.13 |
| $C_{sa}$ | 0.20 | $C_{sa}$ | 0.18 | $C_{sa}$ | 0.17 | $C_{sa}$ | 0.15 | $C_{sa}$ | 0.13 | $C_{sa}$ | 0.11 |
| $E_{max,lv}$ | 0.10 | $E_{max,lv}$ | 0.09 | $E_{max,lv}$ | 0.09 | $E_{max,lv}$ | 0.09 | $E_{max,lv}$ | 0.08 | $E_{shift,ra}$ | 0.10 |
| $E_{shift,ra}$ | 0.05 | $E_{shift,ra}$ | 0.06 | $E_{shift,ra}$ | 0.06 | $\tau_{es,lv}$ | 0.05 | $E_{shift,ra}$ | 0.05 | $E_{max,lv}$ | 0.07 |
| $E_{min,lv}$ | 0.03 | $E_{min,lv}$ | 0.03 | $E_{min,lv}$ | 0.03 | $E_{shift,ra}$ | 0.05 | $\tau_{es,lv}$ | 0.05 | $\tau_{es,lv}$ | 0.04 |
| $E_{max,rv}$ | 0.02 | $E_{max,rv}$ | 0.02 | $C_{pa}$ | 0.03 | $E_{min,lv}$ | 0.03 | $E_{min,lv}$ | 0.05 | $E_{min,lv}$ | 0.03 |
| $V_{0,lv}$ | 0.02 | $V_{0,lv}$ | 0.01 | $R_{pvb}$ | 0.03 | $E_{max,rv}$ | 0.02 | $E_{max,rv}$ | 0.03 | $E_{max,ra}$ | 0.03 |
| $C_{sv}$ | 0.01 | $C_{sv}$ | 0.01 | $E_{max,rv}$ | 0.02 | $E_{max,la}$ | 0.02 | $\tau_{es,rv}$ | 0.02 | $E_{max,rv}$ | 0.03 |
| $E_{max,la}$ | 0.01 | $E_{max,la}$ | 0.01 | $E_{shift,la}$ | 0.02 | $E_{shift,la}$ | 0.02 | $E_{max,la}$ | 0.02 | $\tau_{es,rv}$ | 0.03 |
| $E_{shift,la}$ | 0.01 | $E_{shift,la}$ | 0.01 | $E_{max,la}$ | 0.02 | $C_{sv}$ | 0.01 | $E_{shift,la}$ | 0.02 | $E_{min,ra}$ | 0.02 |
| $\tau_{es,lv}$ | 0.01 | $C_{pa}$ | 0.01 | $C_{sv}$ | 0.02 | $V_{0,lv}$ | 0.01 | $C_{sv}$ | 0.02 | $C_{sv}$ | 0.02 |
| $R_{pvb}$ | 0.01 | $R_{pvb}$ | 0.01 | $V_{0,lv}$ | 0.01 | $C_{pa}$ | 0.01 | $E_{min,rv}$ | 0.01 | $E_{shift,la}$ | 0.02 |
| $E_{min,rv}$ | 0.01 | $E_{min,rv}$ | 0.01 | $E_{min,rv}$ | 0.01 | $E_{min,rv}$ | 0.01 | $V_{0,lv}$ | 0.01 | $E_{max,la}$ | 0.02 |
| $\tau_{es,lv}$ | 0.01 | $\tau_{es,lv}$ | 0.01 | $E_{max,ra}$ | 0.01 | $R_{pvb}$ | 0.01 | $C_{pa}$ | 0.01 | $E_{min,rv}$ | 0.01 |
| | | | | | | $R_{pvb}$ | 0.01 | $R_{pvb}$ | 0.01 | $R_{ra}$ | 0.01 |
| | | | | | | | | | | $C_{pa}$ | 0.01 |
| | | | | | | | | | | $R_{pvb}$ | 0.01 |

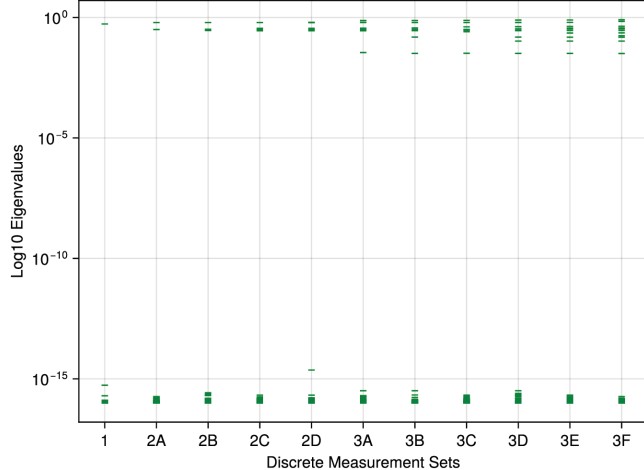

**Fig 4. Discrete measures - sloppy analysis: The eigenvalues of the Fisher information matrix for increasing discrete measurements are displayed on a $\log_{10}$ scale.**

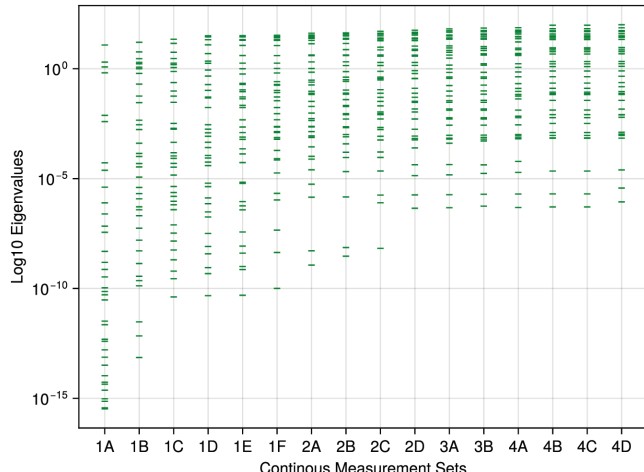

**Fig 5. Continuous measures - sloppy analysis: The eigenvalues of the Fisher information matrix for increasing continuous measurements are displayed on a** $\log_{10}$ **scale.**

observed in the discrete measurement set, when more input parameters are regarded as influential, the concentration of influence decreases and is more evenly spread between the input parameters with an influence value greater than 0.01.

Fig 5 shows that for any continuous measurement set, the system can be regarded as sloppy. When increasing the output measurements, the set of sloppiness decreases, with the eigenvalue spectra decreasing from a range of $10^{-16}$ to $10^{-6}$. Compared to the discrete sloppy analysis (Fig 4), input parameters in the stiff direction exhibit larger values than observed previously.

## Mixed measures

When combing both discrete and continuous measurements, the results in Table 7 show a similar structure to that observed when only continuous measurements are utilised. When only BPN is the only output, all input parameters record an influence greater than 0.01, despite the noise, there are still clear influence parameters ($C_{sa}$, $R_{svb}$ and $\tau_{es,lv}$) which could be regarded as biomarkers. However, when we introduce the noise free BP, the influence values associated with the biomarkers increase largely. At set 4A, the first continuous measurement is introduced alongside the previous discrete ones, as a consequence we observe the number of input parameters with influence greater than 0.01 grows from 9 in set 3D to 17 in set 4A. In set 7D which contains all discrete and continuous measurements, the exact same ranking as set 4D in Table 6 displays, although the values of influence vary slightly. This indicates that continuous measurements dominate, when obtaining influential input parameters for a set of measurements. This pattern is also present in Fig 6, where once continuous measurements are introduced to the output list, sloppiness appears and the eigenvalues are greater than $10^0$ with an eigen-spectrum ranging from $10^{-12}$ for set 4B decreasing down to $10^{-6}$ for set 7D. In sets 5A to 5D, the maximum flow in each chamber compartment was added to the optimisation process as a discrete measurement. Despite discrete measurements being more numerous than continuous ones in our analysis, we acknowledge that mathematically, a continuous measurement is, in principle, infinite-dimensional. While clinically these are often considered

**Table 6. Input parameter rankings for continuous measurements: Input parameters (P) and their corresponding influence values (E) calculated using eqn (11), displayed in decreasing order of influence for each continuous measurement set (from Set 1A to Set 4D). The table shows how influence rankings evolve as additional continuous measurements are incorporated. Note, only parameters with influence values exceeding 0.01 are included.**

**Measurement Sets**

| 1A P | E | 1B P | E | 1C P | E | 1D P | E | 1E P | E | 1F P | E |
|---|---|---|---|---|---|---|---|---|---|---|---|
| $E_{min,lv}$ | 0.57 | $E_{min,lv}$ | 0.32 | $\tau_{es,lv}$ | 0.38 | $E_{shift,la}$ | 0.21 | $E_{shift,la}$ | 0.19 | $E_{shift,la}$ | 0.11 |
| $R_{svb}$ | 0.26 | $E_{shift,la}$ | 0.16 | $E_{min,lv}$ | 0.14 | $\tau_{ep,lv}$ | 0.18 | $\tau_{ep,lv}$ | 0.15 | $E_{shift,ra}$ | 0.1 |
| $C_{sa}$ | 0.23 | $R_{svb}$ | 0.16 | $R_{svb}$ | 0.1 | $E_{min,lv}$ | 0.07 | $E_{min,lv}$ | 0.06 | $\tau_{ep,lv}$ | 0.08 |
| $\tau_{es,lv}$ | 0.16 | $C_{pa}$ | 0.14 | $C_{sa}$ | 0.06 | $\tau_{es,lv}$ | 0.06 | $\tau_{es,lv}$ | 0.05 | $E_{min,lv}$ | 0.04 |
| $C_{pv}$ | 0.15 | $C_{sv}$ | 0.12 | $E_{max,lv}$ | 0.06 | $R_{svb}$ | 0.03 | $\tau_{es,rv}$ | 0.05 | $\tau_{es,rv}$ | 0.04 |
| $E_{shift,la}$ | 0.13 | $C_{sa}$ | 0.11 | $E_{shift,la}$ | 0.05 | $C_{pa}$ | 0.03 | $R_{svb}$ | 0.03 | $\tau_{es,lv}$ | 0.03 |
| $E_{max,lv}$ | 0.13 | $R_{pvb}$ | 0.11 | $C_{sv}$ | 0.04 | $E_{min,la}$ | 0.03 | $C_{pa}$ | 0.03 | $\tau_{ep,rv}$ | 0.03 |
| $C_{sv}$ | 0.12 | $\tau_{es,rv}$ | 0.1 | $C_{pa}$ | 0.04 | $C_{sv}$ | 0.02 | $E_{min,la}$ | 0.02 | $R_{svb}$ | 0.02 |
| $E_{shift,ra}$ | 0.08 | $C_{pv}$ | 0.1 | $C_{pv}$ | 0.04 | $E_{max,la}$ | 0.02 | $C_{sv}$ | 0.02 | $C_{pa}$ | 0.02 |
| $E_{max,la}$ | 0.08 | $E_{max,lv}$ | 0.09 | $R_{pvb}$ | 0.03 | $R_{pvb}$ | 0.02 | $E_{max,la}$ | 0.02 | $C_{sv}$ | 0.02 |
| $E_{min,rv}$ | 0.08 | $\tau_{es,lv}$ | 0.09 | $E_{shift,ra}$ | 0.03 | $\tau_{es,rv}$ | 0.02 | $R_{pvb}$ | 0.02 | $R_{pvb}$ | 0.01 |
| $E_{max,rv}$ | 0.07 | $E_{shift,ra}$ | 0.07 | $\tau_{es,rv}$ | 0.03 | $C_{sa}$ | 0.01 | $C_{sa}$ | 0.02 | $E_{min,la}$ | 0.01 |
| $C_{pa}$ | 0.07 | $E_{min,rv}$ | 0.07 | $E_{min,rv}$ | 0.03 | $E_{max,lv}$ | 0.01 | $E_{max,lv}$ | 0.02 | $E_{min,rv}$ | 0.01 |
| $R_{pvb}$ | 0.04 | $E_{max,la}$ | 0.06 | $\tau_{ep,lv}$ | 0.02 | $C_{pv}$ | 0.01 | $C_{pv}$ | 0.01 | $E_{max,la}$ | 0.01 |
| $E_{max,ra}$ | 0.03 | $E_{max,rv}$ | 0.05 | $E_{max,la}$ | 0.02 | $E_{shift,ra}$ | 0.01 | $E_{min,rv}$ | 0.01 | $E_{max,lv}$ | 0.01 |
| $R_{sv}$ | 0.02 | $\tau_{ep,lv}$ | 0.04 | | | $E_{min,rv}$ | 0.01 | $E_{shift,ra}$ | 0.01 | $C_{sa}$ | 0.01 |
| $E_{min,la}$ | 0.02 | $E_{min,la}$ | 0.03 | | | | | | | $C_{pv}$ | 0.01 |
| | | $E_{max,ra}$ | 0.02 | | | | | | | | |

**Measurement Sets**

| 2A P | E | 2B P | E | 2C P | E | 2D P | E | 3A P | E | 3B P | E |
|---|---|---|---|---|---|---|---|---|---|---|---|
| $\tau_{ep,lv}$ | 0.18 | $\tau_{ep,lv}$ | 0.12 | $E_{shift,la}$ | 0.11 | $E_{shift,ra}$ | 0.16 | $\tau_{ep,lv}$ | 0.12 | $E_{min,lv}$ | 0.1 |
| $E_{shift,la}$ | 0.06 | $E_{shift,ra}$ | 0.1 | $\tau_{ep,lv}$ | 0.1 | $E_{shift,la}$ | 0.06 | $E_{min,lv}$ | 0.09 | $\tau_{ep,lv}$ | 0.08 |
| $E_{min,lv}$ | 0.04 | $E_{shift,la}$ | 0.06 | $E_{min,lv}$ | 0.08 | $E_{min,lv}$ | 0.05 | $E_{shift,ra}$ | 0.08 | $R_{svb}$ | 0.08 |
| $E_{shift,ra}$ | 0.04 | $E_{min,lv}$ | 0.05 | $E_{shift,ra}$ | 0.06 | $\tau_{ep,lv}$ | 0.04 | $E_{shift,la}$ | 0.07 | $E_{shift,ra}$ | 0.07 |
| $\tau_{es,lv}$ | 0.03 | $\tau_{es,rv}$ | 0.03 | $E_{min,la}$ | 0.05 | $E_{min,rv}$ | 0.03 | $E_{min,la}$ | 0.04 | $E_{shift,la}$ | 0.07 |
| $E_{max,lv}$ | 0.02 | $\tau_{es,lv}$ | 0.03 | $E_{max,lv}$ | 0.03 | $E_{min,la}$ | 0.03 | $\tau_{es,lv}$ | 0.03 | $\tau_{es,lv}$ | 0.07 |
| $\tau_{es,rv}$ | 0.02 | $\tau_{ep,rv}$ | 0.03 | $\tau_{es,lv}$ | 0.02 | $\tau_{ep,rv}$ | 0.03 | $E_{max,lv}$ | 0.03 | $E_{min,la}$ | 0.04 |
| $R_{svb}$ | 0.02 | $E_{max,lv}$ | 0.02 | $C_{sv}$ | 0.02 | $C_{sv}$ | 0.02 | $C_{sv}$ | 0.02 | $E_{max,lv}$ | 0.03 |
| $C_{sv}$ | 0.02 | $C_{sv}$ | 0.02 | $\tau_{es,rv}$ | 0.02 | $E_{min,ra}$ | 0.02 | $E_{min,rv}$ | 0.02 | $C_{sv}$ | 0.03 |
| $C_{pa}$ | 0.01 | $R_{svb}$ | 0.02 | $R_{svb}$ | 0.02 | $\tau_{es,rv}$ | 0.02 | $R_{svb}$ | 0.02 | $E_{min,rv}$ | 0.03 |
| $C_{sa}$ | 0.01 | $E_{max,rv}$ | 0.02 | $E_{max,la}$ | 0.02 | $E_{max,rv}$ | 0.02 | $E_{max,rv}$ | 0.02 | $C_{sa}$ | 0.03 |
| $E_{min,la}$ | 0.01 | $C_{pa}$ | 0.01 | $E_{max,rv}$ | 0.02 | $E_{max,lv}$ | 0.02 | $C_{pv}$ | 0.02 | $C_{pv}$ | 0.03 |
| $C_{pv}$ | 0.01 | $E_{min,rv}$ | 0.01 | $C_{pa}$ | 0.02 | $R_{svb}$ | 0.02 | $C_{sa}$ | 0.02 | $E_{max,rv}$ | 0.02 |
| $E_{min,rv}$ | 0.01 | $C_{sa}$ | 0.01 | $C_{sa}$ | 0.02 | $E_{max,ra}$ | 0.02 | $\tau_{es,rv}$ | 0.02 | $E_{max,la}$ | 0.02 |
| | | $C_{pv}$ | 0.01 | $C_{pv}$ | 0.02 | $C_{pa}$ | 0.01 | $C_{pa}$ | 0.02 | $C_{pa}$ | 0.02 |
| | | $E_{min,la}$ | 0.01 | $E_{min,rv}$ | 0.02 | $\tau_{es,lv}$ | 0.01 | $E_{max,la}$ | 0.01 | $\tau_{es,rv}$ | 0.02 |
| | | | | $\tau_{ep,rv}$ | 0.01 | $C_{pv}$ | 0.01 | $\tau_{ep,rv}$ | 0.01 | $\tau_{ep,rv}$ | 0.01 |
| | | | | $R_{pvb}$ | 0.01 | $E_{max,la}$ | 0.01 | $E_{min,ra}$ | 0.01 | $E_{min,ra}$ | 0.01 |
| | | | | | | $C_{sa}$ | 0.01 | $E_{max,ra}$ | 0.01 | $E_{max,ra}$ | 0.01 |

**Measurement Sets**

| 4A P | E | 4B P | E | 4C P | E | 4D P | E |
|---|---|---|---|---|---|---|---|
| $E_{min,lv}$ | 0.09 | $E_{min,lv}$ | 0.09 | $E_{shift,ra}$ | 0.18 | $E_{shift,ra}$ | 0.16 |
| $E_{shift,ra}$ | 0.08 | $E_{shift,ra}$ | 0.09 | $E_{min,rv}$ | 0.05 | $E_{min,lv}$ | 0.08 |
| $R_{svb}$ | 0.07 | $R_{svb}$ | 0.07 | $E_{min,lv}$ | 0.05 | $E_{min,rv}$ | 0.05 |
| $\tau_{ep,lv}$ | 0.06 | $\tau_{ep,lv}$ | 0.06 | $R_{svb}$ | 0.04 | $R_{svb}$ | 0.04 |
| $E_{shift,la}$ | 0.06 | $E_{shift,la}$ | 0.05 | $C_{sv}$ | 0.03 | $E_{shift,la}$ | 0.04 |
| $E_{min,rv}$ | 0.04 | $E_{min,rv}$ | 0.04 | $E_{shift,la}$ | 0.03 | $C_{sv}$ | 0.03 |
| $\tau_{es,lv}$ | 0.03 | $C_{sv}$ | 0.04 | $E_{min,ra}$ | 0.03 | $\tau_{ep,lv}$ | 0.03 |
| $E_{min,la}$ | 0.03 | $E_{min,la}$ | 0.03 | $\tau_{ep,rv}$ | 0.03 | $E_{min,la}$ | 0.03 |
| $C_{sv}$ | 0.03 | $\tau_{es,lv}$ | 0.03 | $\tau_{ep,lv}$ | 0.02 | $E_{min,ra}$ | 0.03 |
| $E_{max,lv}$ | 0.03 | $E_{max,lv}$ | 0.03 | $E_{max,rv}$ | 0.02 | $E_{max,rv}$ | 0.03 |
| $\tau_{ep,rv}$ | 0.02 | $\tau_{es,rv}$ | 0.03 | $\tau_{es,rv}$ | 0.02 | $E_{max,lv}$ | 0.02 |
| $\tau_{es,rv}$ | 0.02 | $\tau_{ep,rv}$ | 0.03 | $E_{max,ra}$ | 0.02 | $\tau_{ep,rv}$ | 0.02 |
| $E_{max,rv}$ | 0.02 | $E_{max,rv}$ | 0.02 | $E_{min,la}$ | 0.02 | $\tau_{es,rv}$ | 0.02 |
| $C_{sa}$ | 0.02 | $C_{pv}$ | 0.02 | $E_{max,lv}$ | 0.01 | $C_{pv}$ | 0.02 |
| $C_{pv}$ | 0.01 | $C_{sa}$ | 0.01 | $C_{pv}$ | 0.01 | $E_{max,ra}$ | 0.02 |
| $C_{pa}$ | 0.01 | $C_{pa}$ | 0.01 | $C_{pa}$ | 0.01 | $\tau_{es,lv}$ | 0.02 |
| $E_{min,ra}$ | 0.01 | $R_{pvb}$ | 0.01 | $\tau_{es,lv}$ | 0.01 | $C_{sa}$ | 0.02 |
| $E_{max,la}$ | 0.01 | $E_{min,ra}$ | 0.01 | $R_{pvb}$ | 0.01 | $C_{pa}$ | 0.02 |
| $E_{max,ra}$ | 0.01 | $E_{max,ra}$ | 0.01 | $C_{sa}$ | 0.01 | $R_{pvb}$ | 0.01 |
| | | $E_{max,la}$ | 0.01 | | | $E_{max,la}$ | 0.01 |

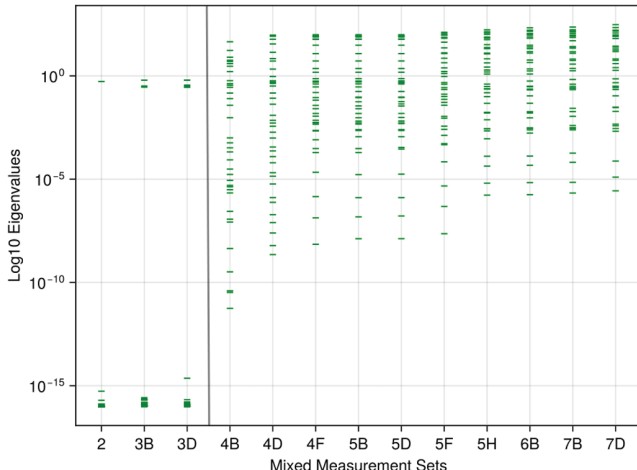

**Fig 6. Mixed measures - sloppy analysis: The eigenvalues of the Fisher information matrix for every other increasing mixed measurements are displayed on a** $\log_{10}$ **scale.** The delineation line between 3D and 4B denotes the stage at which continuous measurements are added to the system.

as independent objects which contain different information, we recognise that continuous functions typically require multi-dimensional representations (unless constant). The addition of these discrete values had no influence on the ranking or sensitivity of the most influential parameters. It is important to note, however, that the maximum value of flow is contained in the continuous measurement, so in effect no new information is provided to the solver; rather, a weighting on the maxima of the flow is applied.

## Discussion

Our study aims to assess the impact of experimental design on the input parameter influence and the system sloppiness. Overall, the results largely agree with previous work: continuous measurements lead to a larger selected subset of input parameters as prime candidates for personalisation in a cardiovascular DT [45,55,58]. When only discrete measurements are used, there is a smaller and more concentrated subset of identifiable input parameters. Perhaps surprising is the quantitative extent of this disparity. Only when the largest discrete measurement set, 3F, is examined, we obtain the same number of input parameters with a value greater than 0.01, compared to the first continuous measurement set 1A. It is important to note that these findings are specifically applicable to the LPM digital twin with Julia implementation in this work. In accordance with established verification and validation (V&V) frameworks, such as ASME V&V40 [85], we acknowledge that computational models inherently contain model uncertainty—uncertainty associated with the structure, assumptions, and limitations of the computational model itself. Without comprehensive code verification and validation procedures, parameter sensitivity results may be influenced by coding errors or flawed hypotheses rather than true physiological relationships. This limitation underscores the necessity of implementing rigorous V&V protocols in future work to distinguish between genuine parameter insensitivity and artefacts introduced by model implementation errors.

We also observe that as the size of the output set increases, the influence between input parameters appears to become more evenly distributed. For example, for the discrete measurements results shown in Table 5, the systemic vascular bed resistance $R_{svb}$ and arterial compliance $C_{sa}$ rank as the most influential parameters for all measurement sets except 2D. In

Sloppy experimental designs in cardiovascular models

**Table 7. Input parameter ranking for mixed measurements: Parameters (P) and their influence values (E) calculated using Eq (11), presented for increasingly complex experimental designs that combine both discrete and continuous measurements.** The progression from Sets 1-7D demonstrates how parameter influence changes when different measurement types are integrated. Only parameters with influence values greater than 0.01 are displayed.

**Measurement Sets**

| 1 P | 1 E | 2 P | 2 E | 3A P | 3A E | 3B P | 3B E | 3C P | 3C E | 3D P | 3D E | 4A P | 4A E | 4B P | 4B E | 4C P | 4C E | 4D P | 4D E | 4E P | 4E E | 4F P | 4F E | 5A P | 5A E |
|---|---|---|---|---|---|---|---|---|---|---|---|---|---|---|---|---|---|---|---|---|---|---|---|---|---|
| $C_{sa}$ | 0.22 | $C_{sa}$ | 0.80 | $C_{sa}$ | 0.44 | $R_{svb}$ | 0.34 | $R_{svb}$ | 0.26 | $E_{shift,ra}$ | 0.24 | $E_{min,lv}$ | 0.44 | $E_{min,lv}$ | 0.27 | $\tau_{es,lv}$ | 0.38 | $\tau_{ep,lv}$ | 0.25 | $\tau_{ep,lv}$ | 0.20 | $\tau_{ep,lv}$ | 0.11 | $\tau_{ep,lv}$ | 0.11 |
| $R_{svb}$ | 0.21 | $R_{svb}$ | 0.60 | $R_{svb}$ | 0.43 | $C_{sa}$ | 0.33 | $C_{sa}$ | 0.26 | $R_{svb}$ | 0.10 | $R_{svb}$ | 0.29 | $C_{pa}$ | 0.16 | $E_{min,lv}$ | 0.11 | $E_{shift,la}$ | 0.16 | $E_{shift,la}$ | 0.14 | $E_{shift,la}$ | 0.09 | $E_{shift,la}$ | 0.09 |
| $\tau_{es,lv}$ | 0.17 | $\tau_{es,lv}$ | 0.04 | $E_{max,lv}$ | 0.24 | $E_{max,lv}$ | 0.18 | $E_{max,lv}$ | 0.14 | $C_{sa}$ | 0.10 | $C_{sa}$ | 0.23 | $R_{svb}$ | 0.16 | $R_{svb}$ | 0.09 | $E_{min,lv}$ | 0.06 | $\tau_{es,rv}$ | 0.05 | $E_{shift,ra}$ | 0.09 | $E_{shift,ra}$ | 0.09 |
| | | | | $V_{0,lv}$ | 0.04 | $V_{0,lv}$ | 0.03 | $V_{0,lv}$ | 0.03 | $E_{max,lv}$ | 0.06 | $C_{sv}$ | 0.22 | $E_{shift,la}$ | 0.16 | $C_{sv}$ | 0.06 | $\tau_{es,lv}$ | 0.05 | $E_{min,lv}$ | 0.05 | $\tau_{es,rv}$ | 0.05 | $\tau_{es,rv}$ | 0.05 |
| | | | | $\tau_{es,lv}$ | 0.02 | $E_{max,rv}$ | 0.02 | $E_{max,rv}$ | 0.02 | $E_{max,ra}$ | 0.04 | $\tau_{es,lv}$ | 0.17 | $C_{sv}$ | 0.14 | $E_{max,lv}$ | 0.06 | $C_{pa}$ | 0.03 | $\tau_{es,lv}$ | 0.04 | $E_{min,lv}$ | 0.04 | $E_{min,lv}$ | 0.04 |
| | | | | | | $\tau_{es,lv}$ | 0.02 | $\tau_{es,lv}$ | 0.02 | $\tau_{ep,rv}$ | 0.02 | $E_{shift,ra}$ | 0.13 | $\tau_{es,rv}$ | 0.12 | $C_{sa}$ | 0.05 | $R_{svb}$ | 0.02 | $C_{pa}$ | 0.03 | $\tau_{ep,rv}$ | 0.03 | $\tau_{ep,rv}$ | 0.03 |
| | | | | | | | | | | $E_{min,rv}$ | 0.02 | $E_{max,lv}$ | 0.13 | $R_{pvb}$ | 0.11 | $E_{shift,la}$ | 0.05 | $C_{sv}$ | 0.02 | $R_{svb}$ | 0.02 | $\tau_{es,lv}$ | 0.03 | $\tau_{es,lv}$ | 0.03 |
| | | | | | | | | | | $E_{max,rv}$ | 0.01 | $E_{min,rv}$ | 0.12 | $C_{sa}$ | 0.10 | $C_{pa}$ | 0.04 | $E_{min,la}$ | 0.02 | $C_{sv}$ | 0.02 | $C_{pa}$ | 0.02 | $C_{pa}$ | 0.02 |
| | | | | | | | | | | $V_{0,lv}$ | 0.01 | $C_{pv}$ | 0.11 | $E_{shift,ra}$ | 0.09 | $E_{shift,ra}$ | 0.03 | $\tau_{es,rv}$ | 0.02 | $E_{min,la}$ | 0.02 | $R_{svb}$ | 0.02 | $R_{svb}$ | 0.02 |
| | | | | | | | | | | | | $E_{shift,la}$ | 0.10 | $E_{max,lv}$ | 0.08 | $E_{min,rv}$ | 0.03 | $R_{pvb}$ | 0.02 | $R_{pvb}$ | 0.02 | $C_{sv}$ | 0.02 | $C_{sv}$ | 0.02 |
| | | | | | | | | | | | | $E_{max,rv}$ | 0.08 | $E_{min,rv}$ | 0.08 | $C_{pv}$ | 0.03 | $E_{max,la}$ | 0.02 | $E_{max,la}$ | 0.02 | $E_{min,rv}$ | 0.01 | $E_{min,rv}$ | 0.01 |
| | | | | | | | | | | | | $E_{max,la}$ | 0.06 | $C_{pv}$ | 0.08 | $R_{pvb}$ | 0.03 | $E_{max,lv}$ | 0.01 | $E_{max,lv}$ | 0.01 | $R_{pvb}$ | 0.01 | $R_{pvb}$ | 0.01 |
| | | | | | | | | | | | | $E_{max,ra}$ | 0.04 | $\tau_{es,lv}$ | 0.07 | $\tau_{es,rv}$ | 0.03 | $E_{shift,ra}$ | 0.01 | $E_{max,rv}$ | 0.01 | $E_{min,la}$ | 0.01 | $E_{min,la}$ | 0.01 |
| | | | | | | | | | | | | $R_{sv}$ | 0.04 | $E_{max,rv}$ | 0.05 | $E_{max,rv}$ | 0.02 | $E_{min,rv}$ | 0.01 | $E_{min,rv}$ | 0.01 | $E_{max,lv}$ | 0.01 | $E_{max,lv}$ | 0.01 |
| | | | | | | | | | | | | $C_{pa}$ | 0.03 | $E_{max,la}$ | 0.04 | $E_{max,la}$ | 0.02 | $C_{sa}$ | 0.01 | $E_{shift,ra}$ | 0.01 | $E_{max,la}$ | 0.01 | $E_{max,la}$ | 0.01 |
| | | | | | | | | | | | | $R_{pvb}$ | 0.03 | $\tau_{ep,lv}$ | 0.03 | $\tau_{ep,lv}$ | 0.02 | $C_{pv}$ | 0.01 | $C_{sa}$ | 0.01 | | | | |
| | | | | | | | | | | | | $E_{min,la}$ | 0.01 | $E_{max,ra}$ | 0.03 | $E_{max,ra}$ | 0.01 | | | $C_{pv}$ | 0.01 | | | | |
| | | | | | | | | | | | | | | $R_{sv}$ | 0.03 | $R_{sv}$ | 0.01 | | | | | | | | |
| | | | | | | | | | | | | | | $E_{min,la}$ | 0.03 | | | | | | | | | | |

(Set 1: All input parameters register $E>0.01$)

**Measurement Sets**

| 5B P | 5B E | 5C P | 5C E | 5D P | 5D E | 5E P | 5E E | 5F P | 5F E | 5G P | 5G E | 5H P | 5H E | 6A P | 6A E | 6B P | 6B E | 7A P | 7A E | 7B P | 7B E | 7C P | 7C E | 7D P | 7D E |
|---|---|---|---|---|---|---|---|---|---|---|---|---|---|---|---|---|---|---|---|---|---|---|---|---|---|
| $\tau_{ep,lv}$ | 0.11 | $\tau_{ep,lv}$ | 0.11 | $\tau_{ep,lv}$ | 0.11 | $\tau_{ep,lv}$ | 0.11 | $\tau_{ep,lv}$ | 0.11 | $E_{shift,la}$ | 0.11 | $E_{shift,ra}$ | 0.16 | $\tau_{ep,lv}$ | 0.16 | $E_{min,lv}$ | 0.12 | $E_{min,lv}$ | 0.09 | $E_{min,lv}$ | 0.09 | $E_{shift,ra}$ | 0.18 | $E_{shift,ra}$ | 0.16 |
| $E_{shift,la}$ | 0.09 | $E_{shift,la}$ | 0.09 | $E_{shift,la}$ | 0.09 | $E_{shift,la}$ | 0.09 | $E_{shift,ra}$ | 0.10 | $\tau_{ep,lv}$ | 0.09 | $E_{shift,la}$ | 0.06 | $E_{min,lv}$ | 0.06 | $\tau_{ep,lv}$ | 0.09 | $E_{shift,ra}$ | 0.09 | $E_{shift,ra}$ | 0.09 | $E_{min,rv}$ | 0.05 | $E_{min,lv}$ | 0.08 |
| $E_{shift,ra}$ | 0.09 | $E_{shift,ra}$ | 0.09 | $E_{shift,ra}$ | 0.09 | $E_{min,lv}$ | 0.05 | $E_{shift,la}$ | 0.06 | $E_{min,lv}$ | 0.07 | $E_{min,lv}$ | 0.05 | $E_{shift,ra}$ | 0.05 | $R_{svb}$ | 0.08 | $R_{svb}$ | 0.07 | $R_{svb}$ | 0.07 | $E_{min,lv}$ | 0.05 | $E_{min,rv}$ | 0.05 |
| $\tau_{es,rv}$ | 0.05 | $\tau_{es,rv}$ | 0.05 | $\tau_{es,rv}$ | 0.05 | $E_{shift,ra}$ | 0.04 | $E_{min,lv}$ | 0.05 | $E_{shift,ra}$ | 0.06 | $\tau_{ep,lv}$ | 0.04 | $E_{shift,la}$ | 0.04 | $E_{shift,la}$ | 0.07 | $\tau_{ep,lv}$ | 0.06 | $\tau_{ep,lv}$ | 0.06 | $R_{svb}$ | 0.04 | $R_{svb}$ | 0.04 |
| $E_{min,lv}$ | 0.04 | $E_{min,lv}$ | 0.04 | $E_{min,lv}$ | 0.04 | $\tau_{es,lv}$ | 0.03 | $\tau_{es,rv}$ | 0.03 | $E_{min,la}$ | 0.05 | $E_{min,rv}$ | 0.03 | $E_{min,la}$ | 0.03 | $E_{min,la}$ | 0.04 | $E_{shift,la}$ | 0.06 | $E_{shift,la}$ | 0.06 | $E_{shift,la}$ | 0.03 | $E_{shift,la}$ | 0.04 |
| $\tau_{ep,rv}$ | 0.03 | $\tau_{ep,rv}$ | 0.03 | $\tau_{ep,rv}$ | 0.03 | $\tau_{es,rv}$ | 0.02 | $\tau_{ep,rv}$ | 0.03 | $E_{max,lv}$ | 0.03 | $E_{min,la}$ | 0.03 | $\tau_{es,lv}$ | 0.03 | $\tau_{es,lv}$ | 0.03 | $E_{min,rv}$ | 0.04 | $E_{min,rv}$ | 0.04 | $C_{sv}$ | 0.03 | $C_{sv}$ | 0.03 |
| $\tau_{es,lv}$ | 0.03 | $\tau_{es,lv}$ | 0.03 | $\tau_{es,lv}$ | 0.03 | $E_{max,lv}$ | 0.02 | $\tau_{es,lv}$ | 0.03 | $\tau_{es,lv}$ | 0.03 | $\tau_{ep,rv}$ | 0.03 | $E_{max,lv}$ | 0.03 | $E_{max,lv}$ | 0.03 | $\tau_{es,lv}$ | 0.03 | $\tau_{es,lv}$ | 0.03 | $E_{min,la}$ | 0.03 | $\tau_{ep,lv}$ | 0.03 |
| $C_{pa}$ | 0.02 | $C_{pa}$ | 0.02 | $C_{pa}$ | 0.02 | $R_{svb}$ | 0.02 | $E_{max,lv}$ | 0.02 | $C_{sv}$ | 0.03 | $C_{sv}$ | 0.03 | $C_{sv}$ | 0.03 | $C_{sv}$ | 0.03 | $E_{min,la}$ | 0.03 | $E_{min,la}$ | 0.03 | $\tau_{ep,rv}$ | 0.02 | $E_{min,la}$ | 0.03 |
| $R_{svb}$ | 0.02 | $R_{svb}$ | 0.02 | $R_{svb}$ | 0.02 | $C_{sv}$ | 0.01 | $R_{svb}$ | 0.02 | $\tau_{es,rv}$ | 0.02 | $E_{min,ra}$ | 0.02 | $E_{min,rv}$ | 0.02 | $E_{min,rv}$ | 0.02 | $C_{sv}$ | 0.03 | $C_{sv}$ | 0.03 | $\tau_{es,lv}$ | 0.02 | $E_{min,ra}$ | 0.02 |
| $C_{sv}$ | 0.02 | $C_{sv}$ | 0.02 | $C_{sv}$ | 0.02 | $C_{pa}$ | 0.01 | $C_{sv}$ | 0.02 | $R_{svb}$ | 0.02 | $\tau_{es,rv}$ | 0.02 | $R_{svb}$ | 0.02 | $R_{svb}$ | 0.02 | $E_{min,ra}$ | 0.02 | $E_{max,lv}$ | 0.03 | $E_{max,rv}$ | 0.02 | $E_{max,rv}$ | 0.02 |
| $E_{min,rv}$ | 0.01 | $E_{min,rv}$ | 0.01 | $E_{min,rv}$ | 0.01 | $C_{sa}$ | 0.01 | $E_{max,rv}$ | 0.02 | $E_{max,rv}$ | 0.02 | $E_{max,rv}$ | 0.02 | $E_{max,rv}$ | 0.02 | $E_{max,rv}$ | 0.02 | $E_{max,rv}$ | 0.02 | $E_{max,rv}$ | 0.03 | $\tau_{es,rv}$ | 0.02 | $E_{max,lv}$ | 0.02 |
| $R_{pvb}$ | 0.01 | $R_{pvb}$ | 0.01 | $R_{pvb}$ | 0.01 | $E_{min,la}$ | 0.01 | $C_{pa}$ | 0.02 | $C_{pa}$ | 0.02 | $E_{max,la}$ | 0.02 | $\tau_{es,rv}$ | 0.02 | $\tau_{es,rv}$ | 0.02 | $\tau_{es,rv}$ | 0.02 | $\tau_{es,rv}$ | 0.03 | $E_{max,rv}$ | 0.02 | $\tau_{ep,rv}$ | 0.02 |
| $E_{min,la}$ | 0.01 | $E_{min,la}$ | 0.01 | $E_{min,la}$ | 0.01 | $C_{pv}$ | 0.01 | $E_{min,rv}$ | 0.02 | $C_{sa}$ | 0.02 | $R_{svb}$ | 0.02 | $C_{pv}$ | 0.02 | $C_{pv}$ | 0.02 | $E_{max,rv}$ | 0.02 | $E_{max,ra}$ | 0.02 | $E_{max,la}$ | 0.02 | $\tau_{es,rv}$ | 0.02 |
| $E_{max,lv}$ | 0.01 | $E_{max,lv}$ | 0.01 | $E_{max,lv}$ | 0.01 | | | $C_{sa}$ | 0.01 | $C_{pv}$ | 0.01 | $E_{max,ra}$ | 0.02 | $C_{sa}$ | 0.02 | $C_{sa}$ | 0.02 | $C_{sa}$ | 0.02 | $E_{max,rv}$ | 0.02 | $E_{max,lv}$ | 0.02 | $E_{max,rv}$ | 0.02 |
| $E_{max,la}$ | 0.01 | $E_{max,la}$ | 0.01 | $E_{max,la}$ | 0.01 | | | $C_{pv}$ | 0.01 | $E_{min,rv}$ | 0.01 | $\tau_{es,lv}$ | 0.02 | $E_{max,la}$ | 0.02 | $E_{max,la}$ | 0.02 | $C_{pv}$ | 0.02 | $\tau_{ep,rv}$ | 0.02 | $C_{pv}$ | 0.01 | $E_{max,la}$ | 0.02 |
| | | | | | | | | $E_{min,la}$ | 0.01 | $\tau_{ep,rv}$ | 0.01 | $C_{pa}$ | 0.02 | $C_{pa}$ | 0.02 | $\tau_{ep,rv}$ | 0.01 | $C_{pa}$ | 0.02 | $E_{max,ra}$ | 0.02 | $C_{pa}$ | 0.01 | $E_{max,la}$ | 0.02 |
| | | | | | | | | | | $R_{pvb}$ | 0.01 | $C_{pv}$ | 0.01 | $\tau_{ep,rv}$ | 0.01 | $E_{min,ra}$ | 0.01 | $C_{pv}$ | 0.02 | $C_{pv}$ | 0.02 | $\tau_{es,lv}$ | 0.01 | $C_{pv}$ | 0.01 |
| | | | | | | | | | | | | $E_{max,la}$ | 0.01 | $E_{min,ra}$ | 0.01 | $E_{max,ra}$ | 0.01 | $C_{sa}$ | 0.02 | $C_{sa}$ | 0.02 | $R_{pvb}$ | 0.01 | $\tau_{es,lv}$ | 0.01 |
| | | | | | | | | | | | | $C_{sa}$ | 0.01 | $E_{max,ra}$ | 0.01 | | | $E_{min,ra}$ | 0.01 | $C_{pa}$ | 0.02 | $C_{sa}$ | 0.01 | $C_{sa}$ | 0.02 |
| | | | | | | | | | | | | | | | | | | $E_{max,ra}$ | 0.01 | $R_{pvb}$ | 0.02 | | | $C_{pa}$ | 0.02 |
| | | | | | | | | | | | | | | | | | | | | $E_{min,ra}$ | 0.01 | | | $R_{pvb}$ | 0.01 |
| | | | | | | | | | | | | | | | | | | | | $E_{max,ra}$ | 0.01 | | | $E_{max,la}$ | 0.01 |

set 1, $C_{sa}$ and $R_{svb}$ have influence values of 0.8 and 0.6 respectively. However, in set 3F, when the dimensionality of output space increases to 11, these two parameters' influence values decrease to 0.11 and 0.13.

The first sloppy analysis of this cardiovascular model indicates that discrete measurements do not introduce sloppiness into the system, whereas for continuous measurements, the system begins to exhibit sloppy behaviour. Through the perspective of creating DTs, the stiff input parameters are clearly identified using discrete measurements which would lead to easier identification of a personalised global minimum parameter set. When using the combination of both the continuous and discrete measurement sets, as they increase in size, the number of stiff input parameters which can be considered as prime candidates for personalisation increases. Sloppiness provides a view into the structure of the input parameter space and an insight for why more "influential" input parameters appear when the dimensionality of the output grows. As more measurements are added, there is a noticeable change in the structure of the response surface, providing more guidance towards the personalisable global minimum. It should be noted that as more and more continuous signals are added, more outputs of the model are constrained. Therefore the increasing number of equally significant parameters is unsurprising. Whilst potential cross-correlation amongst measurements might seem a plausible explanation for this phenomenon, our observations suggest otherwise; parameter significance changes align with the physiological subsystems being measured rather than reflecting mere statistical redundancy, indicating genuine additional model constraint. For specific use cases, this suggests that, where multiple data are available, a weighting of the most important clinical features may be required in the optimisation function to identify the required biomarkers.

When creating a virtual representation of a patient, it is still an open question whether the DT should be personalised to a specific condition or encapsulate the full physiological envelope of a patient [2,10,23]. Our experiments and analysis provides an insight to this question: if one wishes to capture a full physiological envelope through the DT, a number of continuous measurements are essential. This is due to the larger number of influential input parameters, along with the higher values of stiff eigenvalues, when compared to the discrete measurement setting. This approach brings practical problems of course, because of the invasive nature associated with obtaining some continuous measurements (for example, ventricular pressure). A patient would have to be subject to a series of invasive tests with associated risks, in order to generate the data to for a personalised DT. Alongside this, continuous measurements have shown a higher set of sloppiness, indicating that a computationally expensive optimisation routine may have to be utilised to generate the virtual patient representation. Conversely, if the purpose of a DT is to target specific conditions, a set of non-invasive discrete measurements poses as an alternative. Although there is a smaller number of identifiable input parameters in this case, the influence is concentrated strongly around the biomarkers relevant to the discrete metrics. In addition, because the system does not exhibit sloppy behaviour, the personalisation process using discrete measurements may be more efficient than its continuous counterparts. Additionally, continuous measurements taken clinically are also susceptible to noise from several sources, including equipment accuracy and differences in method between operators. In the case of time series data, noise can be present at varying sets at each time step, and therefore it is expected that noise in continuous data will have an increased effect on model sloppiness and parameter identifiability.

One problem with sloppy analysis is the subjectivity in diagnosing whether sloppiness is present within a system. In this work, we have used the common definition of evenly spaced eigenvalues on a logarithmic scale, distributed over a minimum range of 6 orders of magnitude [61,63]. The absence of sloppiness is evident in the discrete measurements setting (see

Fig 4). In the case of continuous measurements, as the measurement set increases, the distribution of eigenvalues (Fig 5), whilst still evenly spaced, reduces from a spread of $10^{16}$ to just $10^6$ (i.e., more input parameters align with the stiff direction than before). But does this apparent reduction in sloppiness align with intuition? Given the increase of parameters in the stiff direction, one would expect more accurate optimisation of the input parameters when compared to set 1. However, this remains an open question and requires further study to investigate. This discrepancy highlights a fundamental challenge in sloppy analysis: the quantitative metrics we employ may not always capture the qualitative behaviour that practitioners intuitively expect from the system.

The study of sloppiness is common practice in most other areas of systems biology, however, this is not the case for cardiovascular models. The concept of sloppiness provides an important insight for examining the personalisability of cardiovascular models. By assessing the stiff and sloppy directions generated from the input parameters, sloppy analysis provides an alternative approach to identify optimal subsets for personalisation, compared to other methods such as profile likelihood and combining sensitivity and orthogonality [58,79,86]. This is an interesting area which should be explored in future research. When attempting to personalise a DT, there are several stages and sloppy analysis belongs to the vital off-line stage in which prime candidates for personalisation are identified. This off-line stage enables us to identify biomarkers for which can be personalised to produce the virtual representation of a patient. The off-line stage is vital because once patient data are introduced, any additional issues occurring during personalisation can then be attributed to issues within the clinical data. If we examine the parameters in Figs 4–6 we observe that the rankings obtained by the analysing their influence (Tables 5–7) are the same rankings obtained through the sloppy analysis. So in the discrete setting the systemic vascular bed resistance $R_{svb}$ and the arterial compliance $C_{sa}$ exhibit the largest eigenvalues making them the stiffest parameters in that setting. Thus these parameters are prime candidates for personalisation.

For the personalisation of cardiovascular DTs, the process in which this happens must operate on a multi-dimensional input parameter space in which some points give accurate representation of a patient's physiological and pathophysiological state. Currently, much analysis on the input parameter space and the identification of the optimal parameter subset for personalisation are conducted on a local basis [25,87,88]. For example, it is still the norm to form the sensitivity matrices through local methods when analysing system sloppiness [35,36]. If sloppy analysis is to be utilised more in the identification of biomarkers, local analysis should not be adopted for larger, more complex circulatory models. Personalisation is a global process, therefore it is vital to understand and quantify the global behaviour and the structure present within the input parameter space. This is why we have conducted our sloppiness analysis using the global sensitivity analysis outcomes in this work. Our workflow could be enhanced by incorporating alternative physiologically-informed priors. When certain parameter combinations produce unrealistic outputs (such as cardiac output or pressures outside clinically accepted ranges), constraints based on population data could be implemented. By leveraging known physiological distributions of outputs from clinical cohorts, we could work backwards to infer suitable distributions of input parameters that would generate these observed output patterns. This Bayesian approach would allow sampling from a more realistic parameter space, potentially improving the efficiency of the personalisation process. Such physiologically-constrained priors would be particularly valuable when working with continuous measurement sets that induce system sloppiness, as they would help narrow the solution space towards clinically relevant parameter combinations. This refinement represents a natural extension of our workflow that maintains the global sensitivity analysis framework while incorporating domain knowledge about cardiovascular physiology. In this work, we

have avoided such a task due to the bias which would be introduced into the input space when trying to identify such physiological regions. This bias is likely to break the orthogonality conditions which are present within the calculation of Sobol indices [89,90]. We will consider and investigate this in future work.

In order to perform such an extensive study, the associated computational expense is another important factor to consider. In total, we have tested 48 individual experimental designs, for each of which a sensitivity analysis has been performed with 75,000 samples to ensure convergence [75]. As our chosen global sensitivity method is Sobol indices [25,53,79], this means for each experimental design, $75000 \times (36 + 2) = 2.85 \times 10^6$ model evaluations are required. Thus for 48 independent experimental designs, we have solved the lumped parameter 4-chamber model 136.8 million times. This study has only been made feasible due to the superior computational speed exhibited by DifferentialEquations.jl within Julia in solving the dynamical system for a single model run including 30 cycles took 0.060246(s). Another approach would be to utilise emulation which is also likely to significantly improve the computational time [46]. However, when emulating such a model, a smooth assumption on the input space is applied which can make identifying unphysiological or unstable regions challenging. When personalising DTs, computationally efficient and accurate tools should be utilised where possible, for the most effective allocation of computational resources for all stages of DT development. Of further interest in the development of LPMs in clinical use is the ability to validate the results of the model. Validation must be suitable for the context of use [91], and therefore also plays a key factor in experimental design, defining the clinical measurements taken and the choice of model output. Experimental design is therefore constricted not only by model behaviour as analysed here, but by clinical requirements. The requirement for testing in the context of use is one of the limitations of general cardiovascular models, as each use case must be included in the validation.

Alongside LPMs, there is also extensive research in higher dimensional (e.g. 1D, 2D and 3D) cardiovascular models which can be utilised as DTs [9,92,93]. The set of physiological details in these models is usually far superior to what can be established in LPMs. The main drawback or compromise, is the lack of ability to simulate global haemodynamics because of the astronomical computational cost. If we were able to create a full cardiovascular circulation representation, which could adapt to pathophysiological states, we would then be able to observe and predict other circulatory diseases, on top of the one in which the current condition occurs. Physiologically detailed models of a single piece of vessel or a compartment are of course of great importance, to further biological understanding where invasive clinical assessments are inappropriate or unethical. One promising area of the cardiovascular digital twin development is in the creation of multi-scale, multi-modal models, combining both LPMs and physiologically detailed higher dimensional representations of specific vessels or valves [46,94]. This approach combines the advantages of both modelling domains and forms an attractive avenue for future research in cardiovascular personalisation and building DTs.

## Conclusion

Our study highlights the importance of the experimental design for the quantification of input parameter influence and the associated sloppiness, for a lumped parameter personalised cardiovascular digital twin. Using a realistic lumped 4-chamber 36-parameter LPM as a test bench, we investigated 48 independent experimental designs. The most significant findings, corresponding to the ones identified in Section Rationale and Contributions, are: (i) Input parameter identifiability is not consistent when subject to varied measurement data and depends on the chosen experimental design. (ii) Sloppiness is present in LPMs, when the

chosen experimental design contains continuous measurements. (iii) The personalisation of a digital twin to encompass a person's complete physiological envelope necessitates invasive tests to obtain continuous measurements. Although this approach offers an increased number of identifiable parameters with potentials to be biomarkers, it comes at the expense of a sloppy system which in turn increases the difficulty in parameter identification. Conversely, discrete metrics may provide a simpler personalisation approach, yielding less identifiable but more targeted biomarkers, due to the absence of sloppiness in the system.

## Author contributions

**Conceptualization:** Harry Saxton, Daniel J. Taylor, Ian Halliday, Xu Xu.

**Formal analysis:** Harry Saxton, Xu Xu.

**Funding acquisition:** Ian Halliday, Torsten Schenkel, Xu Xu.

**Methodology:** Harry Saxton, Daniel J. Taylor, Ian Halliday, Tom Newman, Xu Xu.

**Software:** Torsten Schenkel.

**Supervision:** Ian Halliday, Torsten Schenkel, Richard H. Clayton, Xu Xu.

**Validation:** Ian Halliday, Xu Xu.

**Visualization:** Harry Saxton, Grace Faulkner.

**Writing – original draft:** Harry Saxton, Daniel J. Taylor, Ian Halliday, Torsten Schenkel, Xu Xu.

**Writing – review & editing:** Grace Faulkner, Ian Halliday, Paul D. Morris, Richard H. Clayton, Xu Xu.

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
