## [Decision Letter · Decision Letter 0]

5 Jun 2025

PONE-D-25-06403THE IMPACT OF EXPERIMENTAL DESIGNS & SYSTEM SLOPPINESS ON THE PERSONALISATION PROCESS: A CARDIOVASCULAR PERSPECTIVEPLOS ONE

Dear Dr. Saxton,

Thank you for submitting your manuscript to PLOS ONE. After careful consideration, we feel that it has merit but does not fully meet PLOS ONE’s publication criteria as it currently stands. Therefore, we invite you to submit a revised version of the manuscript that addresses the points raised during the review process.

We look forward to receiving your revised manuscript.

Kind regards,

Pan Li, PhD

Academic Editor

PLOS ONE

**Journal Requirements:**

1. When submitting your revision, we need you to address these additional requirements. Please ensure that your manuscript meets PLOS ONE's style requirements, including those for file naming. The PLOS ONE style templates can be found at https://journals.plos.org/plosone/s/file?id=wjVg/PLOSOne_formatting_sample_main_body.pdf and https://journals.plos.org/plosone/s/file?id=ba62/PLOSOne_formatting_sample_title_authors_affiliations.pdf 2. Please update your submission to use the PLOS LaTeX template. The template and more information on our requirements for LaTeX submissions can be found at http://journals.plos.org/plosone/s/latex. 3. Thank you for stating the following in the Acknowledgments Section of your manuscript: CVD-Net, We gratefully acknowledge Polish high-performance computing infrastructure PLGrid (HPC Center: ACK Cyfronet AGH) for providing computer facilities and support within computational grant no. PLG/2024/017108. We note that you have provided funding information that is not currently declared in your Funding Statement. However, funding information should not appear in the Acknowledgments section or other areas of your manuscript. We will only publish funding information present in the Funding Statement section of the online submission form. Please remove any funding-related text from the manuscript and let us know how you would like to update your Funding Statement. Currently, your Funding Statement reads as follows: The author(s) received no specific funding for this work.  Please include your amended statements within your cover letter; we will change the online submission form on your behalf. 4. Thank you for uploading your study's underlying data set. Unfortunately, the repository you have noted in your Data Availability statement does not qualify as an acceptable data repository according to PLOS's standards. At this time, please upload the minimal data set necessary to replicate your study's findings to a stable, public repository (such as figshare or Dryad) and provide us with the relevant URLs, DOIs, or accession numbers that may be used to access these data. For a list of recommended repositories and additional information on PLOS standards for data deposition, please see https://journals.plos.org/plosone/s/recommended-repositories. 5. When completing the data availability statement of the submission form, you indicated that you will make your data available on acceptance. We strongly recommend all authors decide on a data sharing plan before acceptance, as the process can be lengthy and hold up publication timelines. Please note that, though access restrictions are acceptable now, your entire data will need to be made freely accessible if your manuscript is accepted for publication. This policy applies to all data except where public deposition would breach compliance with the protocol approved by your research ethics board. If you are unable to adhere to our open data policy, please kindly revise your statement to explain your reasoning and we will seek the editor's input on an exemption. Please be assured that, once you have provided your new statement, the assessment of your exemption will not hold up the peer review process.

**Additional Editor Comments:**

Please note that the complete list of comments from one of the reviewers is included in the attachment.

Reviewers' comments:

Reviewer's Responses to Questions

**Comments to the Author**

1. Is the manuscript technically sound, and do the data support the conclusions?

Reviewer #1: Yes

Reviewer #2: Yes

2. Has the statistical analysis been performed appropriately and rigorously? 

Reviewer #1: N/A

Reviewer #2: Yes

3. Have the authors made all data underlying the findings in their manuscript fully available?

Reviewer #1: Yes

Reviewer #2: Yes

4. Is the manuscript presented in an intelligible fashion and written in standard English?

Reviewer #1: Yes

Reviewer #2: Yes

5. Review Comments to the Author

**Reviewer #1:** Saxton et al. explore the personalization process of a simplified cardiovascular model. They introduce the concept of sloppiness, which refers to the geometry of the loss function along the direction of each parameter. They also discuss experimental design, gradually increasing both the amount and type of data (from discrete to continuous) across experiments. The results show that experimental design affects parameter identifiability (and how parameters are ranked) and that continuous data increases sloppiness, leading to greater numerical difficulty in fitting the data.

The study is both interesting and timely. Several groups are currently developing cardiovascular digital twins and facing similar challenges to those highlighted in this manuscript. The topic is well-suited for PLOS ONE. Overall, the manuscript is well-written, with only a few minor typographical errors. However, the figures could be improved (see comments below).

Major Points:

- Section 2.1.3: Sloppiness certainly affects the choice of numerical strategy (in fact, in numerical analysis, it is related to the concept of “stiffness”). However, the key issue here is different: a sloppy system tends to have elongated or even flat valleys around the minimum, meaning that the minimum is not well-defined. As a result, even small noise in the data could lead to a large region of valid parameter values.

- Prior distribution of parameters: The choice of prior distributions is crucial for improving identifiability, especially in low-data regimes. Here, the prior is likely a uniform distribution (though this is not explicitly stated). While a uniform prior is valid in the absence of additional information, the authors should discuss how alternative priors could be incorporated into their workflow. For instance, certain parameter combinations may produce unrealistic physiological outputs (e.g., cardiac output or pressures outside expected ranges). Given population-based distributions of outputs, one could attempt to infer a suitable distribution of input parameters such that random sampling yields the observed output distribution.

- Experimental design strategy: My understanding is that each experiment, associated with different measurement sets, is independent of the others. However, suppose we start with the smallest measurement set and fit the parameters: could this information be used to optimize the selection of the next measurement set? Certain parameter combinations may exhibit low sloppiness, which could guide the experimental design.

- Section 3.2: Is the clinical data real or synthetically generated? What do you mean by “medically accurate”? Please clarify.

- Tables 5 and 6: These tables are not very accessible. Consider replacing them with a plot. For instance, a “rank plot” could effectively illustrate how parameter importance changes.

- Page 16: The sentence “Despite the number of discrete measurements outnumbering the number of continuous measurements” is misleading. A continuous measurement is, in principle, infinite-dimensional. While a function can often be approximated using a low-dimensional representation due to correlation, this representation typically has more than one dimension (unless the function is constant).

- Page 20: The paragraph beginning with “One problem of the sloppy analysis” is unclear. Could you refine and clarify this explanation?

Minor Points:

- Abstract: Why use “reduced-order” in the first sentence?

- Abstract: The term sloppiness is not defined. This concept may not be widely known outside statistical modeling communities. Consider providing a brief explanation.

- Author summary: Is this section a requirement for PLOS ONE?

- Terminology: The model is often referred to as a “4-chamber model.” While technically accurate, the term “4-chamber model” nowadays typically refers to a 3D electromechanical model of the heart. Consider using “lumped model” or a similar term to avoid confusion.

- Section 2.1.3: The loss function can be visualized in dimensions greater than two using various machine-learning techniques. A simple approach would be to slice the loss landscape with randomly chosen 2D planes.

- Page 12: “poreviosu” -> “previous”

- Figure sequence: Figure 5 is discussed before Figure 4 (page 14). Consider reordering the discussion or figures for better flow.

- Figure 6: You might add a delineation line between 3D and 4B and clarify that the addition of continuous data is being illustrated.

- Page 20: “personlisation” -> “personalisation”

**Reviewer #2:** The document “impact of experimental designs & system sloppiness on the personalisation process: a cardiovascular perspective” presents a thorough sensitivity analysis in a simplified model of the cardiovascular system leverages these results to explain the importance on the input parameters depending on the quantities of interest (QoIs) extracted from (and compared with) the physical system. Particularly, the authors are interested in the sloppiness of the system, this is the separation between the impact of parameters in the outputs, which of course change with the outputs. Particularly, the authors discover that, for this particular model, if the researcher wants to compare QoIs that are continuous clinical measures, there is an increase in the system sloppiness, compared to a discrete measurement approach.

While this work represents a good enough body of work for a publication, I consider there are items that could be improved to increase the technical quality and ease its understanding.

6. PLOS authors have the option to publish the peer review history of their article (what does this mean?). If published, this will include your full peer review and any attached files.

Reviewer #1: No

Reviewer #2: **Yes: **Alfonso Santiago

---

## [Author Response · Author response to Decision Letter 1]

14 May 2025

Please see the attached document with the responses.

---

## [Decision Letter · Decision Letter 1]

26 May 2025

THE IMPACT OF EXPERIMENTAL DESIGNS & SYSTEM SLOPPINESS ON THE PERSONALISATION PROCESS: A CARDIOVASCULAR PERSPECTIVE

PONE-D-25-06403R1

Dear Dr. Saxton,

We’re pleased to inform you that your manuscript has been judged scientifically suitable for publication and will be formally accepted for publication once it meets all outstanding technical requirements.

Kind regards,

Pan Li, PhD

Academic Editor

PLOS ONE

Additional Editor Comments (optional):

Reviewers' comments:

Reviewer's Responses to Questions

**Comments to the Author**

1. If the authors have adequately addressed your comments raised in a previous round of review and you feel that this manuscript is now acceptable for publication, you may indicate that here to bypass the “Comments to the Author” section, enter your conflict of interest statement in the “Confidential to Editor” section, and submit your "Accept" recommendation.

Reviewer #1: All comments have been addressed

Reviewer #2: All comments have been addressed

2. Is the manuscript technically sound, and do the data support the conclusions?

Reviewer #1: Yes

Reviewer #2: Yes

3. Has the statistical analysis been performed appropriately and rigorously? 

Reviewer #1: N/A

Reviewer #2: Yes

4. Have the authors made all data underlying the findings in their manuscript fully available?

Reviewer #1: Yes

Reviewer #2: Yes

5. Is the manuscript presented in an intelligible fashion and written in standard English?

Reviewer #1: Yes

Reviewer #2: Yes

6. Review Comments to the Author

Reviewer #1: There are some latex typos I've spotted:

- line 490-491, the sentence is weird

- line 534, fig -> Figure

Reviewer #2: To this reviewer point of view, the author addressed all the concerns pointed out by the reviewing panel, either by modifying the text or by giving a detailed explanation of the reason the authors chose and making a brief comment on the text. From this reviewer’s perspective, the manuscript is approved for publication after some minor edition comments, see below.

Minor edition issues detected during this last review are:

* Some of the latex \ref seem broken (check e.g. tag 3/33 lines 102,107; page 4/33 lines 109, 114,136 page 5/33 line 175 and so on)

* On the new text page 3/33 lines 62-63 there’s a one sentence paragraph. While it’s not explicitly out of format, it’s stylistically weak. I recommend removing the end of paragraph, making it: “(…) or ‘model calibration’). Mathematically (…)”

* Page 11/33 line 349 Oxford comma missing “In tables 2,3!,! and 4,”. Not mandatory but preferred in formal writing.

7. PLOS authors have the option to publish the peer review history of their article (what does this mean?). If published, this will include your full peer review and any attached files.

Reviewer #1: **Yes: **Simone Pezzuto

Reviewer #2: **Yes: **Alfonso Santiago

---

## [Editor Report · Acceptance letter]

PONE-D-25-06403R1

PLOS ONE

Dear Dr. Saxton,

I'm pleased to inform you that your manuscript has been deemed suitable for publication in PLOS ONE. Congratulations! Your manuscript is now being handed over to our production team.

Kind regards,

on behalf of

Dr. Pan Li

Academic Editor

PLOS ONE